# Efficient Training of Sparse Autoencoders for Large Language Models via Layer Groups

## Abstract

Sparse Autoencoders (SAEs) have recently been employed as an unsupervised approach for understanding the inner workings of Large Language Models (LLMs). They reconstruct the model's activations with a sparse linear combination of interpretable features. However, training SAEs is computationally intensive, especially as models grow in size and complexity. To address this challenge, we propose a novel training strategy that reduces the number of trained SAEs from one per layer to one for a given group of contiguous layers. Our experimental results on Pythia 160M highlight a speedup of up to 3x without compromising the reconstruction quality and performance on downstream tasks. Therefore, layer clustering presents an efficient approach to train SAEs in modern LLMs.

## 1 Introduction

With the significant adoption of Large Language Model (LLM)s in world applications, understanding their inner workings has gained paramount importance. A key challenge in LLMs interpretability is the polysemanticity of neurons in models' activations, lacking a clear and unique meaning (Olah et al., 2020). Recently, SAEs (Huben et al., 2024; Bricken et al., 2023) have shown great promise to tackle this problem by decomposing the model's activations into a sparse combination of human-interpretable features.

The use of SAEs as an interpretability tool is motivated by two key reasons: the first is the substantial empirical evidence supporting the *Linear Representation Hypothesis (LRH)*, or that LLMs exhibit interpretable linear directions in their activation space (Mikolov et al., 2013; Nanda et al., 2023; Park et al., 2023); the second is the *Superposition Hypothesis (SH)* (Elhage et al., 2022), which supposes that, by leveraging sparsity, neural networks represent more features than they have neurons. Under this hypothesis, we can consider a trained neural network as a compressed simulation of a larger disentangled model, where every neuron corresponds to a single feature. To overcome superposition, SAEs leverage the LRH to decompose model activations into a sparse linear combination of interpretable features.

However, training SAEs is computationally expensive and will become even more costly as the model size and parameter counts grow. Indeed, one Sparse Autoencoder (SAE) is typically learned for a given component at every layer in a LLM. Moreover, the number of features usually equals the model activation dimension multiplied by a positive integer, called the expansion factor. For example, a single SAE trained on the Llama-3.1 8B model activations with an expansion factor of 32 has roughly $4096^2 \times 32 \times 2 \approx 1.073$ billion parameters, resulting in a total of more than 32 billions parameters when training one for each of the 32 layers. Additionally, every SAE is trained on a fixed number of tokens $T$, typically in the billions, leading to a cumulative training requirement of $L \times T$ tokens, where $L$ is the total number of layers—for the Llama-3.1 8B model, this amounts to 32 billion tokens. This high computational demand not only increases training time but also requires substantial hardware resources and energy consumption, making the approach increasingly impractical as models scale.

In this work, we reduce the computational overhead of training a separate SAE for each layer of a target LLM by learning a single SAE for groups of related and contiguous layers, thus reducing

---

*Equal contribution

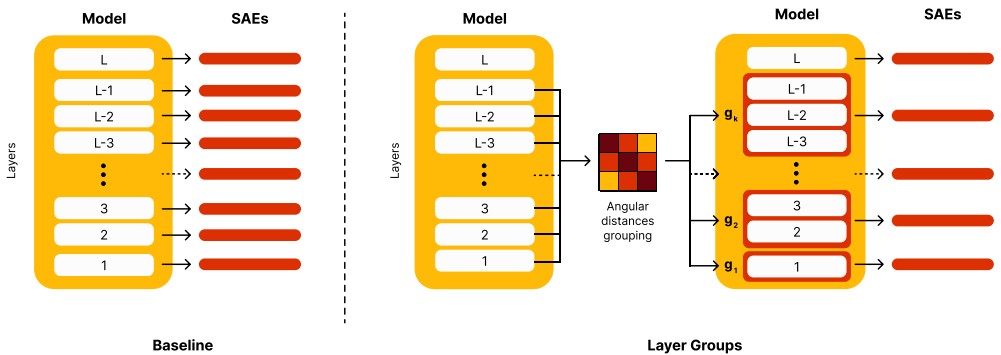

Figure 1: The illustration of our method. While standard training of SAEs requires training one per layer, our method first clusters layers by angular similarity and then trains a single SAE for each group.

the number of trained SAEs from $L$ to $k$, where $k$ is the number of groups. Furthermore, to ensure computational efficiency remains consistent, we fix the total number of training tokens $T$ per group and allocate $T/l$ tokens to each layer within the group, where $l$ is the number of layers in that group. Thus, our approach reduces the total number of training tokens from $LT$ to $kT$. This approach, which we term **Group-SAE**, is inspired by the observation that neural network layers often group together in learning task-specific representations (Szegedy et al., 2014; Zeiler & Fergus, 2014; Jawahar et al., 2019): shallow layers typically capture low-level features, while deeper layers learn high-level abstractions. Additionally, adjacent layers in LLMs tend to encode redundant information, as evidenced by the similarity in the angular distance of their outputs (Gromov et al., 2024). To quantify the relatedness between layers, we measure the angular distance of the residual stream after the MLP's contribution across neighboring layers.

Denoting with $L$ the number of layers of the target model and $k$ the number of groups we want to cluster the layers into, our approach obtains at least a $\frac{(L-1)}{k}$ times speedup[1] without sacrificing neither reconstruction quality nor downstream performance.

Our **contributions** can be summarized as follows:

- We demonstrate that a single Group-SAE can effectively reconstruct the activations of an entire cluster as a sparse linear combination of interpretable features. This approach reduces the number of SAEs required to be trained for a given model by a factor of $k$.

- We demonstrate the practical effectiveness of our method by training SAEs on the Pythia-160M model for different values of $k$, and we obtain a $\frac{L-1}{k}$ times speedup at the cost of a minor deterioration in performance.

- We extensively analyze our method on reconstruction performance, downstream tasks, and human interpretability for different values of $k$.

## 2 RELATED WORK

### 2.1 THE LINEAR REPRESENTATION AND SUPERPOSITION HYPOTHESES

Supported by substantial evidence, from the seminal Mikolov et al. (2013) vector arithmetic to the more recent work of Nanda et al. (2023) and Park et al. (2023) on LLMs, the Linear Representation Hypothesis (LRH) supposes that neural networks have interpretable linear directions in their acti-

---

[1]We do not consider the last transformer layer, as different from every other layer w.r.t. the angular distance defined in Section 3.1.

vations space. However, neuron polysemanticity remains an essential challenge in neural network interpretability (Olah et al., 2020).

Recently, Bricken et al. (2023) explored this issue by relating the Superposition Hypothesis (SH) to the decomposition ideally found by a SAE. According to the SH, neural networks utilize $n$-dimensional activations to encode $m \gg n$ features by leveraging their sparsity and relative importance. As a result, we can write the activations $\mathbf{x}^j$ in a model as

$$\mathbf{x}^j \approx \mathbf{b} + \sum_i^m \boldsymbol{f}_i(\mathbf{x}^j)\mathbf{d}_i \tag{1}$$

where $\mathbf{x}^j \in \mathbb{R}^n$ is the activation vector for an example $j$, $\boldsymbol{f} \in \mathbb{R}^m$ is a sparse feature vector, $\boldsymbol{f}_i(\mathbf{x}^j)$ is the activation of the i-th feature, $\mathbf{d}_i \in \mathbb{R}^n$ is a unit vector in the activation space and $\mathbf{b} \in \mathbb{R}^n$ is a bias term.

## 2.2 SPARSE AUTOENCODERS

Sparse Autoencoders have gained popularity in LLM interpretability due to their ability to counteract superposition and decompose neuron activations into interpretable features (Bricken et al., 2023; Huben et al., 2024). Given an input activation $\mathbf{x} \in \mathbb{R}^{d_{\text{model}}}$, a SAE reconstructs it as a sparse linear combination of $d_{\text{sae}} \gg d_{\text{model}}$ features, denoted as $\mathbf{v}_i \in \mathbb{R}^{d_{\text{model}}}$, where $d_{\text{sae}}$ is set as $d_{\text{sae}} = c \cdot d_{\text{model}}$, where $c \in \{2^n | n \in \mathbb{N}_+\}$. The reconstruction follows the form:

$$(\hat{\mathbf{x}} \circ \mathbf{f})(\mathbf{x}) = \mathbf{W}_d\mathbf{f}(\mathbf{x}) + \mathbf{b}_d \tag{2}$$

Here, the columns of $\mathbf{W}_d \in \mathbb{R}^{d_{\text{model}} \times d_{\text{sae}}}$ represent the features $\mathbf{v}_i$, $\mathbf{b}_d \in \mathbb{R}^{d_{\text{model}}}$ is the decoder's bias term, and $\mathbf{f}(\mathbf{x}) \in \mathbb{R}^{d_{\text{sae}}}$ represents the sparse features activations. The feature activations are computed as

$$\mathbf{f}(\mathbf{x}) = \sigma(\mathbf{W}_e(\mathbf{x} - \mathbf{b}_d) + \mathbf{b}_e) \tag{3}$$

where $\mathbf{W}_e \in \mathbb{R}^{d_{\text{sae}} \times d_{\text{model}}}$ and $\mathbf{b}_e \in \mathbb{R}^{d_{\text{sae}}}$ are the encoder's matrix and bias term respectively, and $\sigma$ is an activation function, typically $\text{ReLU}(\mathbf{x}) = \max(0, \mathbf{x})^2$. The training of a SAE involves minimizing the following loss function:

$$\mathcal{L}_{\text{sae}} = \|\mathbf{x} - \hat{\mathbf{x}}(\mathbf{f}(\mathbf{x}))\|_2^2 + \lambda S(\mathbf{f}(\mathbf{x})) \tag{4}$$

where the first term in Equation 4 represents the reconstruction error, while the second term S is the regularization on the activations $\mathbf{f}(\mathbf{x})$ to encourage sparsity, which is typically equal to the $\ell_1$ norm of the features f(x), i.e. $S(\mathbf{f}(\mathbf{x})) = \|\mathbf{f}(\mathbf{x})\|_1$ Bricken et al. (2023). Other definitions of sparsity are admissible: Rajamanoharan et al. (2024) set S equal to the $\ell_0$ norm of the features, i.e. $S(\mathbf{f}(\mathbf{x})) = \|\mathbf{f}(\mathbf{x})\|_0 = \sum_{i=0}^{d_{\text{sae}}} \mathbb{I}[\boldsymbol{f}_i(\mathbf{x}) \neq 0]$, which is the one adopted in this work along with the JumpReLU activation function, while Gao et al. (2024) imposes sparsity by selecting the Top-K activating features from f(x).

## 2.3 SAEs EVALUATION

SAE evaluation in the context of LLMs presents a significant challenge. While standard unsupervised metrics such as $L_2$ (reconstruction) loss and $L_0$ sparsity are widely adopted to measure SAE performance (Gao et al., 2024; Lieberum et al., 2024), they fall short of assessing two key aspects: causal importance and interpretability.

Recent approaches, including auto-interpretability (Bricken et al., 2023; Huben et al., 2024; Bills et al., 2023) and ground-truth comparisons (Sharkey et al., 2023), aim to provide a more holistic evaluation. These methods focus on the causal relevance of features (Marks et al., 2024) and evaluate SAEs in downstream tasks. Makelov et al. (2024), for instance, proposed a framework for the

---

[2]Other activation functions are admissible, e.g. Top-K (Gao et al., 2024) or JumpReLU (Rajamanoharan et al., 2024), with JumpReLU being the one employed in this work.

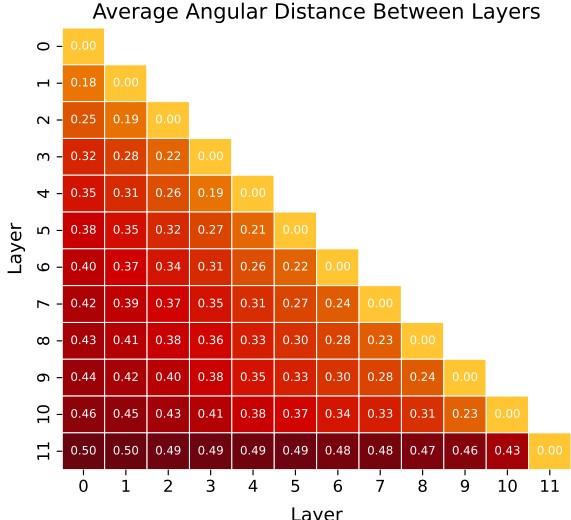

Figure 2: Average angular distance between all layers of the Pythia-160M model, as defined in Equation 5. The angular distances are computed over 5M tokens from the training dataset. The angular distances are bounded in $[0, 1]$, where an angular distance equal to $0$ means equal activations, $0.5$ means activations are perpendicular and an angular distance of $1$ means that the activations point in opposite directions.

Indirect Object Identification (IOI) task, emphasizing three aspects: the sufficiency and necessity of reconstructions, sparse feature steering (Templeton et al., 2024), and the interpretability of features in causal terms.

Karvonen et al. (2024) further contributed by developing specialized metrics for board game language models. Using structured games like chess and Othello, they introduced supervised metrics, such as board reconstruction accuracy and coverage of predefined state properties, offering a more direct assessment of SAEs' ability to capture semantically meaningful and causally relevant features.

## 2.4 IMPROVING SAEs TRAINING

As SAEs gain popularity for LLMs interpretability and are increasingly applied to state-of-the-art models (Lieberum et al., 2024), the need for more efficient training techniques has become evident. To address this, (Gao et al., 2024) explored the scaling laws of Autoencoders to identify the optimal combination of size and sparsity.

Recent work also explored using transfer learning to improve SAE training. For example, Kissane et al. (2024) and Lieberum et al. (2024) demonstrated the transferability of SAE weights between base and instruction-tuned versions of Gemma-1 (Team et al., 2024a) and Gemma-2 (Team et al., 2024b), respectively. On the other hand, Ghilardi et al. (2024) shows that transfer also occurs among layers of a single model, both in forward and backward directions.

## 3 EXPERIMENTAL SETUP

### 3.1 LAYER GROUPS

For a model with $L$ layers, the number of possible combinations of $k$ groups of adjacent layers that can be tested is given by $\binom{L-1}{k-1}$. With this number growing with model depth, we employed an agglomerative grouping strategy based on the angular distances between layers to reduce it drastically. In particular, we compute the mean angular distances, as specified in Gromov et al. (2024), over 5M tokens from our training set:

$$d_{\text{angular}}\left(\mathbf{x}_{\text{post}}^{p}, \mathbf{x}_{\text{post}}^{q}\right) = \frac{1}{\pi} \arccos\left(\frac{\mathbf{x}_{\text{post}}^{p} \cdot \mathbf{x}_{\text{post}}^{q}}{\left\|\mathbf{x}_{\text{post}}^{p}\right\|_{2}\left\|\mathbf{x}_{\text{post}}^{q}\right\|_{2}}\right) \quad (5)$$

for every $p, q \in \{1, ..., L\}$, where $\mathbf{x}_{\text{post}}^{l}$ are the $l$-th residual stream activations after the MLP's contribution[3]. From Figure 8, it can be noted how the last layer is different from every other layer in terms of angular distance. For this reason, we have decided to exclude it from the grouping procedure.

We adopted a bottom-up hierarchical clustering strategy with a complete linkage (Nielsen, 2016) that aggregates layers based on their angular distances. Specifically, at every step of the process, the two groups with minimal group-distance[4] are merged; we repeat the process until the predefined number of groups $k$ is reached. This approach prevents the formation of groups as long chains of layers and ensures that the maximum distance within each group remains minimal. In this work, we create groups considering all layers except the last, and we choose $k$ varying from 1 (a single cluster with all layers) to 5. Layer groups can be found in Appendix B. To show how our method scales to larger models, we also compute distances and groups of Gemma-2 2b and 9b Team et al. (2024b). Detailed results are provided in Appendix B.

## 3.2 SAEs TRAINING AND EVALUATION

We denote $\text{SAE}_i$ as the baseline SAE trained to reconstruct the activations of layer $i$. For every $1 \leq j \leq k \leq 5$ with $j, k \in \mathbb{N}$, we define $\text{SAE}_{j_k}$ as the SAE trained to reconstruct the activations for all layers in the $j$-th group of a partition consisting of $k$ groups. Furthermore, let $[j_k]$ represent the set of layers belonging to the $j$-th group within this partition. Each $\text{SAE}_{j_k}$ is specifically trained to reconstruct the activations of each layer $s \in [j_k]$ individually, using the formulations described in equations 3 and 2.

To ensure both computational efficiency and fair comparison with baselines, we allocate a fixed total of 1 billion tokens for training both every $\text{SAE}_i$ and $\text{SAE}_{j_k}$. In particular, for $\text{SAE}_{j_k}$ these tokens are evenly distributed across the layers in $[j_k]$, assigning $1\text{B}/|[j_k]|$ tokens to each layer, with a randomly chosen layer's activation per token. This approach ensures balanced training across layers within the group while keeping the total token budget consistent with the baseline configurations, thus reducing the total number of training tokens from $LT$ to $kT$.

For evaluation, we train a $\text{SAE}_{j_k}$ for every combination of $k$ and $j$ and compare its performance with the corresponding baseline $\text{SAE}_s$, where $s \in [j_k]$. Section 4 presents the results of these comparisons based on standard reconstruction and sparsity metrics. Furthermore, in Section 5, we show how our approach performs on popular downstream tasks (e.g., (Marks et al., 2024; Hanna et al., 2023; Wang et al., 2023)), while Section 6 reports the human interpretability scores.

## 3.3 DATASET AND HYPERPARAMETERS

We train SAEs on the residual stream of the Pythia-160M model (Biderman et al., 2023) for 1B tokens, focusing on the residual stream activations after the MLP contribution. The chosen dataset is a 2B pre-tokenized version[5] of the Pile dataset (Gao et al., 2020) with a context size of 1024. We set the expansion factor $c = 8$, $\lambda = 3e\text{-}4$ in Equation 4, learning rate equal to $7e\text{-}4$, and a batch size of 4096 samples. Following Bricken et al. (2023), we constrain the decoder columns to have unit norm and do not tie the encoder and decoder weights. Additionally, we have normalized the activations so that they have mean squared $\ell_2$ norm of one during SAE training, as specified in Rajamanoharan et al. (2024), estimating the scaling factor over 2M tokens of our train set.

We use the JumpReLU activation function as specified in Rajamanoharan et al. (2024), and defined as $\text{JumpReLU}_{\theta}(z) = z \cdot \text{H}(z - \theta)$, with $\theta$ being a threshold learned during training and H is the

---

[3]In a Transformer model (Vaswani et al., 2017), the residual stream activations are computed as follows: given the activations $\mathbf{x}_{\text{pre}}^{l-1}$ from layer $l-1$, the pre-layer activations for layer $l$, $\mathbf{x}_{\text{pre}}^{l} = \mathbf{x}_{\text{post}}^{l-1}$, are calculated as $\mathbf{x}_{\text{post}}^{l-1} = \mathbf{x}_{\text{mid}}^{l-1} + \text{MLP}(\mathbf{x}_{\text{mid}}^{l-1})$, where $\mathbf{x}_{\text{mid}}^{l-1} = \mathbf{x}_{\text{pre}}^{l-1} + \text{Attn}(\mathbf{x}_{\text{pre}}^{l-1})$. All SAEs are trained specifically on the residual stream after the MLP contribution, i.e., $\mathbf{x}_{\text{post}}^{l}$, for every layer $l$.

[4]The complete linkage clustering strategy defines the group-distance between two groups $X$ and $Y$ as $D(X, Y) = \max_{\mathbf{x} \in X, \mathbf{y} \in Y} d_{\text{angular}}(\mathbf{x}, \mathbf{y})$

[5]https://huggingface.co/datasets/NeelNanda/pile-small-tokenized-2b

Heaviside function. We fixed the hyperparameters for all the experiments conducted in this work. All hyperparameters can be found in Table 2.

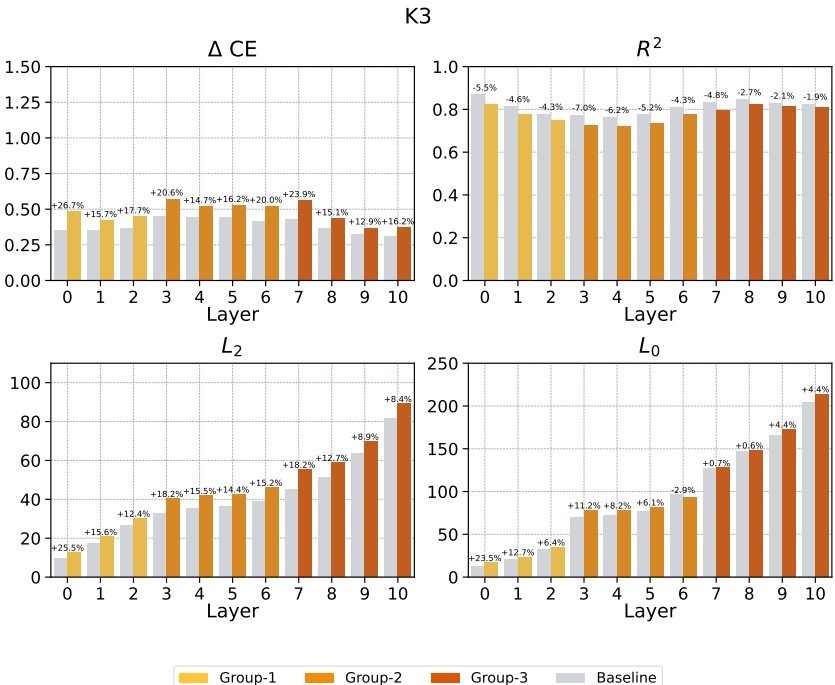

Figure 3: $\Delta$CE, $R^2$, $L_2$ and $L_0$ for $k = 3$ number of groups. Note that, for $\Delta$CE and $L_2$, the lower is the score, the better it is, while the contrary is true for $R^2$. For $L_0$ there are no indications on a preferred value.

## 4 RECONSTRUCTION EVALUATION

To assess the quality of the reconstruction of SAEs trained with our grouping strategy, we report three standard reconstruction metrics and one sparsity metric. In particular, the Delta Cross-Entropy, defined as $\mathrm{CE}(\hat{\mathbf{x}} \circ \mathbf{f}) - \mathrm{CE}(M)$, measures the difference in Cross-Entropy loss (CE) between the output with SAE reconstructed activations ($\hat{\mathbf{x}} \circ \mathbf{f}$) and the model's output ($M$).

The $L_2$ loss is the first term of Equation 4 and measures the reconstruction error made by the SAE. The $R^2$ score, defined as $1 - \|\mathbf{x} - \hat{\mathbf{x}}\|_2^2 / \|\mathbf{x} - \mathbb{E}_{\mathbf{x} \sim \mathcal{D}}[\mathbf{x}]\|_2^2$, measures the fraction of explained variance of the input recovered by the SAE.

Finally, the $L_0$ sparsity loss, defined as $\sum_{j=1}^{d_{\mathrm{sae}}} \mathbb{I}[\mathbf{f}_j \neq 0]$, represents the number of non-zero SAE features used to compute the reconstruction. For each metric, we compute the average over 1M examples from the test dataset.

Figure 3 shows the reconstruction and sparsity metrics for each layer of the grouping with $k = 3$. It can be noted that training a single SAEs on the activations from multiple close layers doesn't dramatically affect the reconstruction, even when all layers are clustered in a small number of groups. Metrics for all other values of $k$ can be found in Appendix C.

These results demonstrate that a single SAE can effectively reconstruct activations across multiple layers. Furthermore, the comparable performance between $\mathrm{SAE}_{j_k}$ and the individual layer-specific $\mathrm{SAE}_i$ indicates that post-residual stream activations from adjacent layers share a common set of underlying features. This hypothesis is further supported by directly comparing the directions learned by $\mathrm{SAE}_i$ and $\mathrm{SAE}_{j_k}$ using the Mean Maximum Cosine Similarity (MMCS) score, as shown in Appendix D.

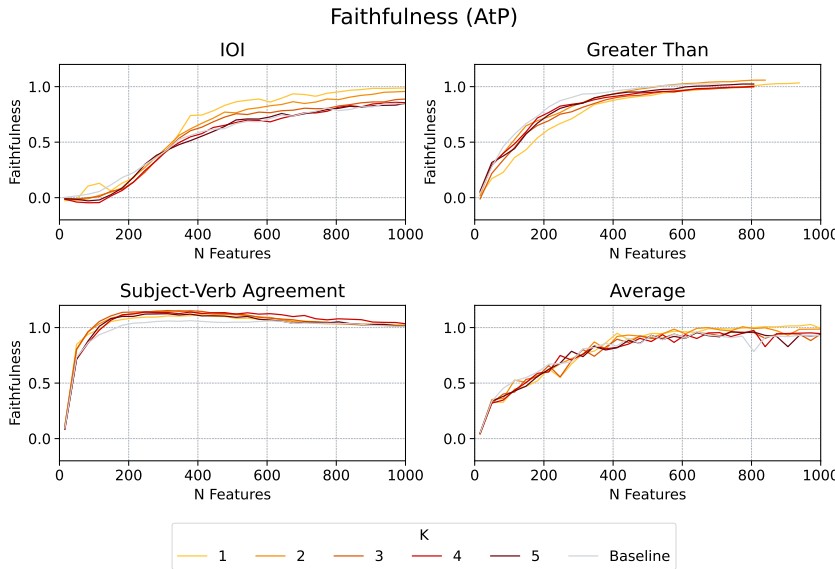

Figure 4: Average faithfulness (Marks et al., 2024) for every downstream task (IOI, Greater Than, Subject-Verb Agreement) with IE computed with AtP (Equation 8). The "Baseline" average is computed considering the performance obtained by $\text{SAE}_i$, $\forall i = 0, ..., 10$. The "Average" plot depicts the average over the three downstream tasks.

## 5 DOWNSTREAM TASKS EVALUATION

While achieving good reconstruction metrics is crucial for a trained SAE, it is insufficient for a comprehensive evaluation of its performance. For instance, unsupervised metrics alone cannot determine whether the identified features capture causally influential directions for the model. To address this, following Marks et al. (2024), we applied SAEs to three well-known tasks: Indirect Object Identification (IOI), Greater Than, and Subject-Verb Agreement.

Each task can be represented as a set of counterfactual prompts paired with their respective answers, formally denoted as $\mathcal{T} : \{x_{\text{clean}}, x_{\text{corrupted}}, a_{\text{clean}}, a_{\text{corrupted}}\}$. Counterfactual prompts are similar to clean ones but contain slight modifications that result in a different predicted answer.

Ideally, a robust SAE should be able to recover the model's performance on a task when reconstructing its activations. Furthermore, we expect the SAE to rely on a small subset of task-relevant features to complete the task. To assess this, we filtered the features to include only the most important ones, where importance is defined as the *indirect effect* (IE) (Pearl, 2022) of the feature on task performance, measured by a real-valued metric $m : \mathbb{R}^{d_{\text{vocab}}} \to \mathbb{R}$. Specifically,

$$\text{IE}(m; \boldsymbol{f}_i; x_{\text{clean}}, x_{\text{corrupted}}) = m(x_{\text{clean}}|\text{do}(\boldsymbol{f}_i = \boldsymbol{f}_{i;\text{corrupted}})) - m(x_{\text{clean}}) \tag{6}$$

In this equation, $\boldsymbol{f}_{i;\text{corrupted}}$ represents the value of the $i$-th feature $\boldsymbol{f}_i$ during the computation of $m(x_{\text{corrupted}})$, and $m(x_{\text{clean}}|\text{do}(\boldsymbol{f}_i = \boldsymbol{f}_{i;\text{corrupted}}))$ refers to the value of $m$ for $x_{\text{clean}}$ under an intervention where the activation of feature $\boldsymbol{f}_i$ is set to $\boldsymbol{f}_{i;\text{corrupted}}$. Moreover, $m$ is defined as the difference in logits between the clean and the corrupted answer.

Calculating these effects is computationally expensive, as it requires a forward pass for each feature. To mitigate this, we employed two approximate methods: Attribution Patching (AtP)(Nanda, 2023; Syed et al., 2024) and Integrated Gradients (IG)(Sundararajan et al., 2017). Appendix E provides a formal definition of both methods.

Following Marks et al. (2024), we used faithfulness and completeness metrics to evaluate the performance of the SAEs on the tasks. These metrics are defined as $\frac{m(C)-m(\emptyset)}{m(M)-m(\emptyset)}$, where $m(M)$ and $m(\emptyset)$ represent the metric average over $\mathcal{T}$, achieved by the model alone and with the mean-ablated

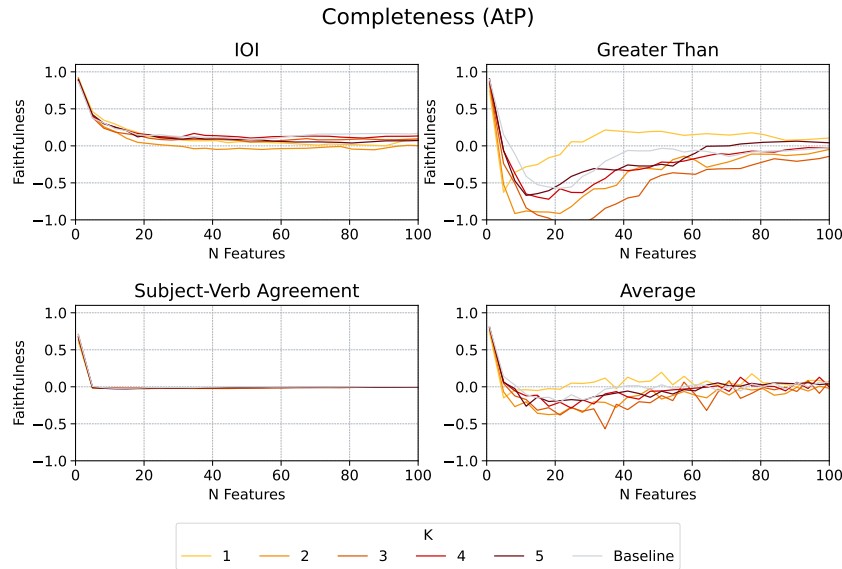

Figure 5: Average completeness for every downstream task (IOI, Greater Than, Subject-Verb Agreement) with IE computed with AtP (Equation 8). The "Baseline" average is computed considering the performance obtained by $\text{SAE}_i$, $\forall i = 0, ..., 10$. The "Average" plot depicts the average performance over the three downstream tasks.

SAE reconstructions, respectively. $m(C)$ is computed based on the task and either the *faithfulness* or *completeness* criteria: for faithfulness, it is the metric average when using only the important SAE features, while mean-ablating the others; on the contrary for completeness, it is calculated by mean-ablating the important features while keeping the others active.

Given a predefined number N of features to compute a circuit, we select the top-N features based on their indirect effect at each layer. Then, to calculate the Faithfulness and Completeness scores, we respectively retain or ablate these selected features while mean-ablating or retaining all the others. As in Marks et al. (2024), the computation of both scores incorporates the SAE error $\epsilon_{\mathbf{x}}$[6] into the reconstructions, as excluding it compromises the model's performance on the task. Furthermore, when using Group-SAEs, the selected features can vary across layers within a group, as each layer may require a distinct set of features to achieve optimal reconstruction, thus $\text{SAE}_{j_k}$ is used to independently and separately reconstruct the activations of every layer $s \in [j_k]$ with the *inderect effect* that is estimated for every layer separately.

These metrics allow us to evaluate two critical aspects of SAE quality: whether the SAE learned a set of features that is both **sufficient** and **necessary** to perform the task. Figures 4 and 5 display the faithfulness and completeness scores for all $k$ groups.

All $\text{SAE}_{j_k}$ perform comparably to, or slightly better than, the baseline $\text{SAE}_i$ models on the faithfulness score (Figure 4) across all three downstream tasks evaluated, demonstrating the sufficiency of learned features. Remarkably, this performance is achieved using only the top 5% of the most important active features, which recover 75% of the baseline performance, on average. Even more remarkable are the performances of the single-group $\text{SAE}_{j_1}$ ($k = 1$), which closely follow the trend of both the baseline SAEs and the $\text{SAE}_{j_5}$ ($k = 5$). Moreover, we did not observe any substantial differences in performance for any of the tested values of $k$. The necessity of the features learned by both the baseline $\text{SAE}_i$ and $\text{SAE}_{j_k}$ is confirmed by the completeness scores depicted in Figure 5, with a severe drop in performance even with only the top 10 active features mean-ablated.

The results of $\text{SAE}_{j_k}$ on downstream tasks demonstrate that their learned features are both sufficient and necessary. Moreover, these findings confirm those in Section 4, i.e., that a single SAE can

---
[6]The SAE error $\epsilon$ is defined as $\epsilon_{\mathbf{x}} = \mathbf{x} - \hat{\mathbf{x}}(\mathbf{f}(\mathbf{x}))$.

effectively reconstruct activations across multiple contiguous layers and learn a set of shared, general features that span adjacent layers.

## 6 HUMAN INTERPRETABILITY

In addition to achieving excellent reconstruction and downstream performance, SAEs must learn interpretable features. Following Ghilardi et al. (2024), we engaged human annotators to identify interpretable patterns in the feature annotations. Specifically, they attempted to provide clear definitions for each feature by examining its top and bottom logits attribution scores, as well as the top activating tokens. In total, we randomly sample 96 features for each SAE and store their activations on 1M tokens from the training dataset. To ensure learned features are not layer-specific, for Group-SAEs we keep the sampled features constant across the layers within the group. To evaluate the quality of these features, we defined the *Human Interpretability Score* as the ratio of features considered interpretable by the human annotators.

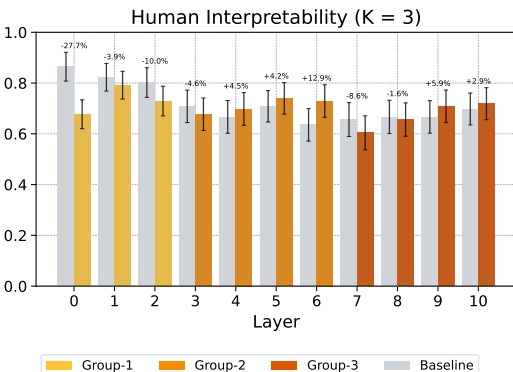

Figure 6: Human Interpretability Scores for $k = 3$. The differences in the interpretability scores of features learned by the $\text{SAE}_{j_k}$ and the baseline $\text{SAE}_i$ are not statistically significant different for all the layers except for the first. Error bars shows one standard deviations of the scores differences, modeled as a Binomial distribution (Wasserman, 2010).

Figure 6 presents the Human Interpretability Scores for all layers of $k = 3$. Scores for all other values of $k$ can be found in Appendix F. According to human annotators, the interpretability of the features learned by $\text{SAE}_{j_k}$ is comparable to that of the baseline $\text{SAE}_i$. Moreover, we found that when many layers are grouped together, e.g., when $k = 1$, $\text{SAE}_{j_k}$ features are more polysemantic overall, probably due to increased interference. Nevertheless, some of them remain perfectly interpretable across all model layers, capturing critical directions in model computations.

## 7 CONCLUSION

This work introduces a novel approach to efficiently train Sparse Autoencoders (SAEs) for Large Language Models (LLMs) by clustering layers based on their angular distance and training a single SAE for each group. Through this method, we achieved up to a 3x speedup in training without compromising reconstruction quality or performance on downstream tasks. The results demonstrate that activations from adjacent layers in LLMs share common features, enabling effective reconstruction with fewer SAEs.

Our findings also show that the SAEs trained on grouped layers perform comparably to layer-specific SAEs in terms of reconstruction metrics, faithfulness, and completeness on various downstream tasks. Furthermore, human evaluations confirmed the interpretability of the features learned by our SAEs, underscoring their utility in disentangling neural activations.

The methodology proposed in this paper opens avenues for more scalable interpretability tools, facilitating deeper analysis of LLMs as they grow in size. Future work will focus on further optimizing the number of layer groups and scaling the approach to even larger models.

## 8 LIMITATIONS AND FUTURE WORKS

One limitation of our approach is the absence of a precise method for selecting the optimal number of layer groups ($k$). This choice is due to the lack of a clear *elbow rule* for identifying the correct number of groups.

Based on our results on Pythia-160m, Appendix B provides the Maximum Average Angular Distance (MAAD) within groups for Pythia-160m and larger models (Gemma-2 2b and 9b). This heuristic can guide the selection of the number of groups $k$ for clustering layers in larger models, relying on the assumption that angular distance between activations from different layers is an important indicator of how the Group SAE will perform on them. While further investigation is needed to refine group configurations and assess scalability, these heuristics offer a practical reference for training Group-SAEs in larger models.

Additionally, we tested our method primarily on the Pythia-160M model, a relatively small LLM. While our findings demonstrate significant improvements in efficiency without sacrificing performance, the scalability of our approach to much larger models remains an open question. Future work could explore how the grouping strategy and training techniques generalize to models with billions of parameters, where the computational benefits would be even more pronounced.

Another important direction for future research involves understanding how Sparse Autoencoders (SAEs) handle the superposition hypothesis when encoding information from multiple layers. While our method effectively grouped layers and maintained high performance, how SAEs manage the potential overlap in feature representation across layers remains unclear. Investigating this aspect could lead to a more clear understanding of the trade-offs between sparsity and feature disentanglement in SAEs, and inform strategies for improving interpretability without compromising task performance.

In summary, while our work represents an efficient step forward in training SAEs for interpretability, extending this approach to larger models and exploring the handling of superposition will provide valuable insights for both practical applications and the theoretical understanding of sparse neural representations.

## 9 REPRODUCIBILITY STATEMENT

To support the replication of our empirical findings on training SAEs via layer groups and to enable further research on understanding their inner works, we plan to release all the code and SAEs used in this study upon acceptance.

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

# A  HYPERPARAMETERS

Table 1: Pythia-160M model specifics

| Config | Value |
|---|---|
| Layers ($L$) | 12 |
| Model dimension ($d_{\text{model}}$) | 768 |
| Heads ($H$) | 12 |
| Non-Embedding params | 85,056,000 |
| Equivalent models | GPT-Neo OPT-125M |

Table 2: Training and fine-tuning hyperparameters

| Hyperparameter | Value |
|---|---|
| c | 8 |
| $\lambda$ | 3e-4 |
| Hook name | resid-post |
| Batch size | 4096 |
| Adam ($\beta_1, \beta_2$) | $(0, 0.999)$ |
| Context size | 1024 |
| lr | 7e-4 |
| lr scheduler | constant |
| lr deacy steps | last 20% of the training steps |
| l1 warm-up steps | 5% of the training steps |
| # tokens (Train) | 1B |
| Checkpoint freq | 200M |
| Decoder weights initialization | Zeroes |
| Activation function | JumpReLU |
| Decoder column normalization | Yes |
| Activation normalization | Mean squared $\ell_2$ norm of one during SAE training |
| FP precision | 32 |
| Prepend BOS token | No |

# B   ADDITIONAL ANGULAR DISTANCES AND LAYERS GROUPS

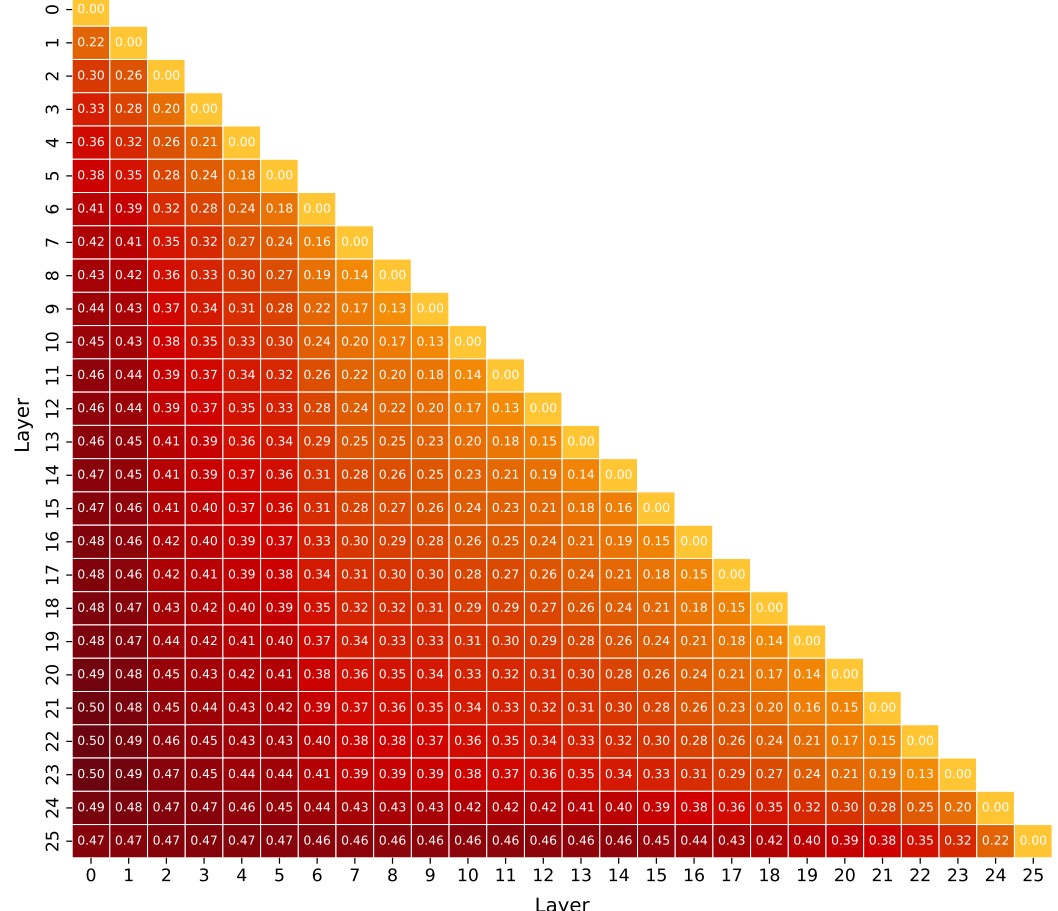

Figure 7: Average angular distance between all layers of Gemma-2 2b, as defined in Section 3.1. The angular distances are bounded in $[0, 1]$, where an angular distance equal to $0$ means equal activations, $0.5$ means activations are perpendicular and an angular distance of $1$ means that the activations point in opposite directions.

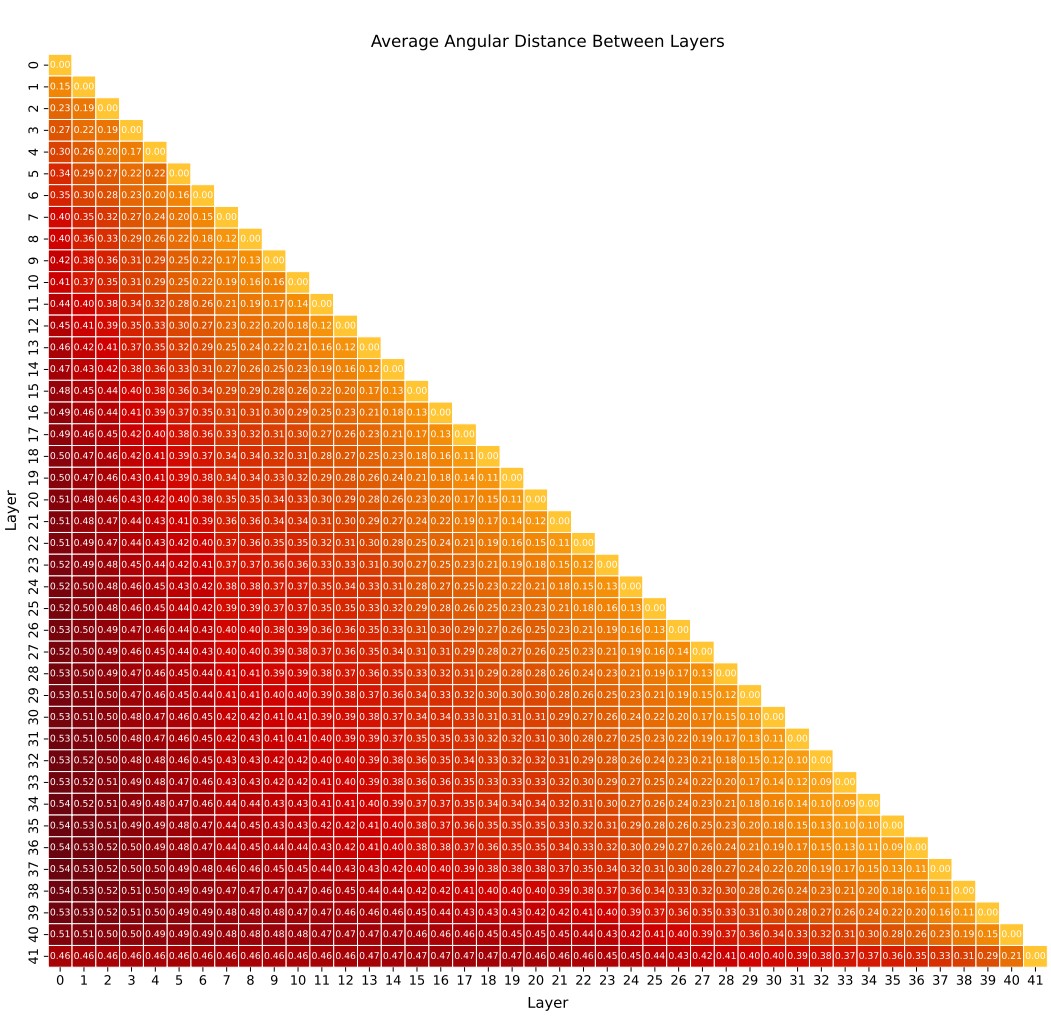

Figure 8: Average angular distance between all layers of Gemma-2 9b, as defined in Section 3.1. The angular distances are bounded in $[0, 1]$, where an angular distance equal to $0$ means equal activations, $0.5$ means activations are perpendicular and an angular distance of $1$ means that the activations point in opposite directions.

| k | Groups | MAAD |
|---|--------|------|
| 2 | 0, 0, 0, 0, 0, 0, 0, 1, 1, 1, 1 | 0.389 |
| 3 | 0, 0, 0, 1, 1, 1, 1, 2, 2, 2, 2 | 0.332 |
| 4 | 0, 0, 0, 1, 1, 1, 1, 2, 2, 3, 3 | 0.311 |
| 5 | 0, 0, 0, 1, 1, 2, 2, 3, 3, 4, 4 | 0.252 |

Table 3: Layer groups for every $k$ up to $L/2$ for Pythia-160m

| k | Groups | MAAD |
|---|--------|------|
| 2 | 0, 0, 0, 0, 0, 0, 0, 0, 0, 0, 0, 0, 0, 0, 0, 0, 0, 0, 1, 1, 1, 1, 1, 1, 1, 1 | 0.476 |
| 3 | 0, 0, 0, 0, 0, 0, 1, 1, 1, 1, 1, 1, 1, 1, 1, 1, 1, 1, 2, 2, 2, 2, 2, 2, 2, 2 | 0.384 |
| 4 | 0, 0, 0, 0, 0, 0, 1, 1, 1, 1, 1, 1, 1, 1, 1, 1, 1, 1, 2, 2, 2, 2, 2, 2, 3, 3 | 0.384 |
| 5 | 0, 0, 0, 0, 0, 0, 1, 1, 1, 1, 1, 1, 1, 1, 1, 2, 2, 2, 2, 2, 2, 3, 3, 3, 4, 4 | 0.384 |
| 6 | 0, 0, 1, 1, 1, 1, 2, 2, 2, 2, 3, 3, 3, 3, 3, 3, 3, 4, 4, 4, 4, 4, 4, 5, 5 | 0.279 |
| 7 | 0, 0, 1, 1, 1, 1, 2, 2, 2, 2, 3, 3, 3, 3, 4, 4, 4, 5, 5, 5, 5, 5, 5, 6, 6 | 0.279 |
| 8 | 0, 0, 1, 1, 1, 1, 2, 2, 2, 2, 3, 3, 3, 3, 4, 4, 4, 5, 5, 5, 5, 6, 6, 7, 7 | 0.279 |
| 9 | 0, 0, 1, 1, 2, 2, 3, 3, 3, 3, 3, 4, 4, 4, 4, 5, 5, 5, 6, 6, 6, 6, 7, 7, 8, 8 | 0.238 |
| 10 | 0, 0, 1, 1, 2, 2, 3, 3, 4, 4, 5, 5, 5, 5, 6, 6, 6, 7, 7, 7, 7, 8, 8, 9, 9 | 0.222 |

Table 4: Layer groups for every $k$ up to $L/2$ for Gemma-2 2b

| k | Groups | MAAD |
|---|--------|------|
| 2 | 0, 0, 0, 0, 0, 0, 0, 0, 0, 0, 0, 0, 0, 0, 0, 0, 0, 0, 0, 0, 0, 0, 0, 0, 1, 1, 1, 1, 1, 1, 1, 1, 1, 1, 1, 1, 1, 1, 1 | 0.524 |
| 3 | 0, 0, 0, 0, 0, 0, 0, 0, 0, 0, 0, 0, 0, 0, 0, 1, 1, 1, 1, 1, 1, 1, 1, 1, 1, 1, 2, 2, 2, 2, 2, 2, 2, 2, 2, 2, 2, 2, 2, 2 | 0.467 |
| 4 | 0, 0, 0, 0, 0, 0, 0, 0, 0, 0, 0, 0, 0, 1, 1, 1, 1, 1, 1, 1, 1, 1, 1, 1, 1, 1, 1, 2, 2, 2, 2, 2, 2, 2, 2, 2, 2, 3, 3, 3 | 0.467 |
| 5 | 0, 0, 0, 0, 0, 0, 0, 0, 0, 0, 0, 0, 1, 1, 1, 1, 1, 1, 1, 1, 1, 1, 2, 2, 2, 2, 2, 2, 2, 2, 2, 2, 2, 3, 3, 3, 4, 4, 4, 4 | 0.333 |
| 6 | 0, 0, 0, 0, 0, 0, 0, 0, 0, 0, 1, 1, 1, 1, 1, 1, 1, 1, 1, 1, 2, 2, 2, 2, 2, 2, 3, 3, 3, 4, 4, 4, 4, 4, 5, 5, 5, 5, 5 | 0.333 |
| 7 | 0, 0, 0, 0, 0, 0, 0, 0, 0, 1, 1, 1, 2, 2, 2, 2, 2, 2, 3, 3, 3, 3, 3, 4, 4, 4, 4, 5, 5, 5, 5, 5, 6, 6, 6, 6, 6 | 0.305 |
| 8 | 0, 0, 0, 0, 0, 0, 0, 1, 1, 1, 2, 2, 2, 2, 2, 3, 3, 3, 3, 3, 4, 4, 4, 4, 5, 5, 5, 5, 6, 6, 6, 6, 7, 7, 7, 7 | 0.305 |
| 9 | 0, 0, 0, 1, 1, 1, 1, 2, 2, 2, 2, 2, 3, 3, 3, 3, 3, 4, 4, 4, 4, 5, 5, 5, 5, 5, 6, 6, 6, 6, 6, 7, 7, 7, 7, 8, 8 | 0.305 |
| 10 | 0, 0, 0, 0, 0, 1, 1, 1, 1, 1, 1, 2, 2, 2, 2, 2, 3, 3, 3, 4, 4, 4, 4, 5, 5, 5, 5, 5, 6, 6, 6, 6, 6, 7, 7, 8, 8, 9 | 0.251 |
| 11 | 0, 0, 0, 0, 0, 0, 1, 1, 1, 1, 1, 2, 2, 2, 2, 2, 3, 3, 3, 4, 4, 4, 4, 5, 5, 6, 6, 6, 6, 6, 7, 7, 8, 8, 9, 10, 10 | 0.227 |
| 12 | 0, 0, 0, 0, 0, 0, 1, 1, 1, 2, 2, 2, 2, 3, 3, 3, 4, 4, 4, 4, 5, 5, 6, 6, 6, 6, 6, 7, 7, 8, 8, 9, 10, 10, 11, 11 | 0.216 |
| 13 | 1, 1, 1, 1, 2, 2, 2, 2, 2, 3, 3, 3, 4, 4, 4, 4, 5, 5, 6, 6, 6, 7, 7, 8, 8, 9, 10, 10, 11, 11, 12, 12 | 0.206 |
| 14 | 0, 0, 0, 0, 0, 0, 1, 1, 1, 1, 2, 2, 2, 2, 3, 3, 3, 3, 4, 4, 4, 4, 5, 5, 6, 6, 7, 7, 8, 8, 9, 10, 10, 11, 11, 12, 12, 13, 13, 13 | 0.202 |
| 15 | 0, 0, 0, 0, 1, 1, 1, 1, 1, 2, 2, 2, 2, 3, 3, 3, 3, 4, 4, 5, 5, 5, 6, 6, 7, 7, 8, 8, 9, 10, 10, 11, 11, 12, 12, 13, 13, 13, 14, 14 | 0.202 |
| 16 | 0, 0, 0, 0, 0, 0, 1, 1, 1, 1, 2, 2, 2, 2, 3, 3, 4, 4, 4, 5, 5, 5, 6, 6, 6, 7, 7, 8, 8, 9, 10, 10, 11, 11, 12, 12, 13, 13, 13, 14, 14, 15, 15 | 0.202 |
| 17 | 0, 0, 0, 0, 0, 1, 1, 2, 2, 2, 3, 3, 4, 4, 5, 5, 5, 6, 6, 6, 7, 7, 8, 8, 9, 10, 10, 11, 11, 12, 12, 13, 13, 13, 14, 14, 15, 15, 16, 16 | 0.202 |
| 18 | 0, 0, 0, 0, 1, 1, 2, 2, 2, 3, 3, 4, 4, 4, 5, 5, 6, 6, 6, 7, 7, 8, 8, 8, 9, 10, 10, 11, 11, 12, 12, 13, 13, 13, 14, 14, 15, 15, 16, 16, 17, 17 | 0.202 |
| 19 | 0, 0, 0, 1, 1, 2, 2, 2, 3, 3, 4, 4, 4, 5, 5, 6, 6, 6, 7, 7, 8, 8, 8, 9, 10, 10, 11, 11, 12, 12, 13, 13, 14, 14, 15, 15, 16, 16, 17, 17, 18, 18 | 0.175 |
| 20 | 0, 0, 1, 1, 1, 2, 2, 2, 3, 3, 4, 4, 5, 5, 6, 6, 6, 7, 7, 8, 8, 8, 9, 10, 10, 11, 11, 12, 12, 13, 13, 14, 14, 15, 15, 16, 16, 17, 17, 18, 18, 19 | 0.175 |

Table 5: Layer groups for every $k$ up to $L/2$ for Gemma-2 9b

## C  DETAILED PER-LAYER RECONSTRUCTION AND SPARSITY PLOTS

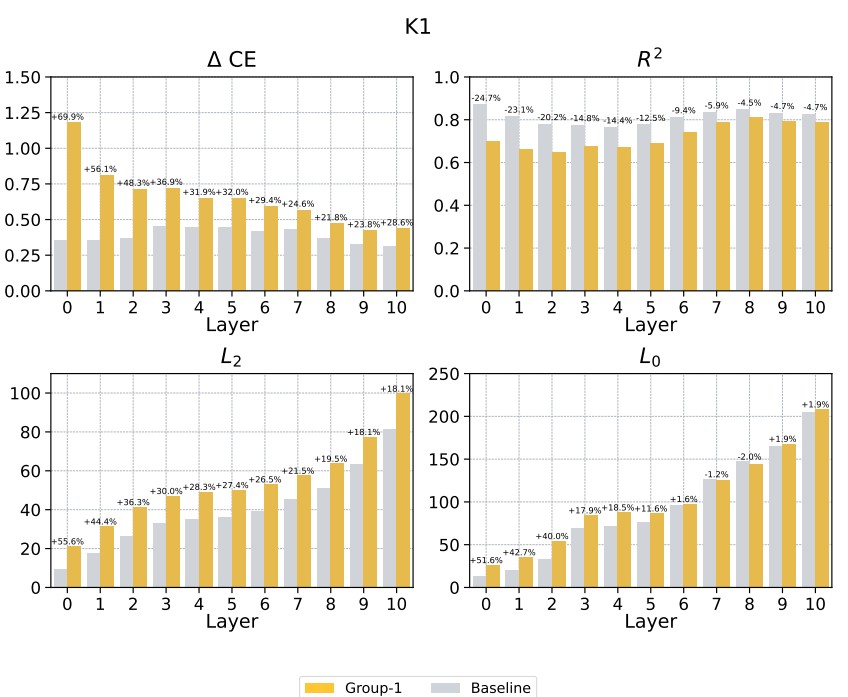

Figure 9: Per-layer $\Delta$CE, $R^2$, $L_2$ and $L_0$ with $k = 1$.

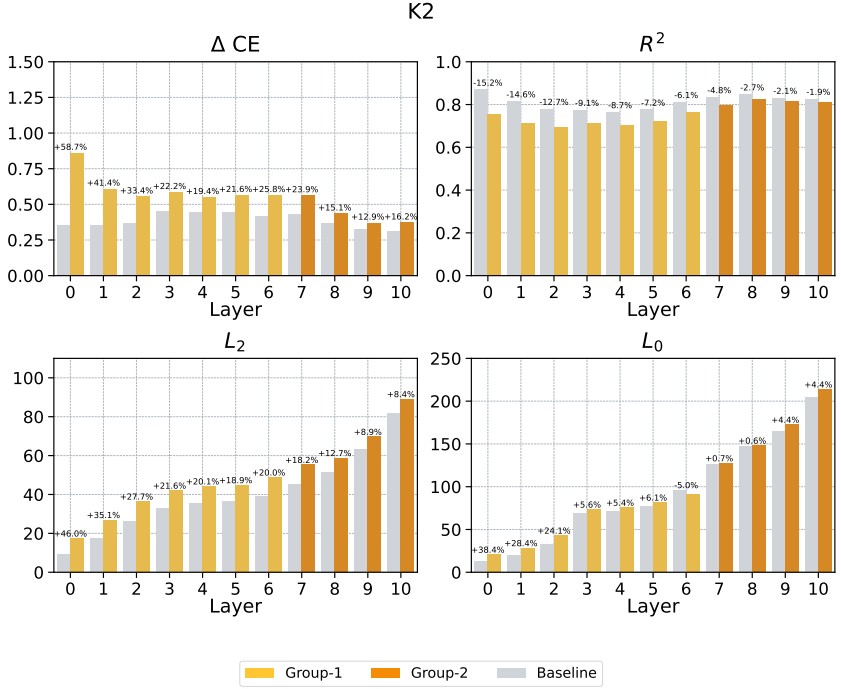

Figure 10: Per-layer $\Delta$CE, $R^2$, $L_2$ and $L_0$ with $k = 2$.

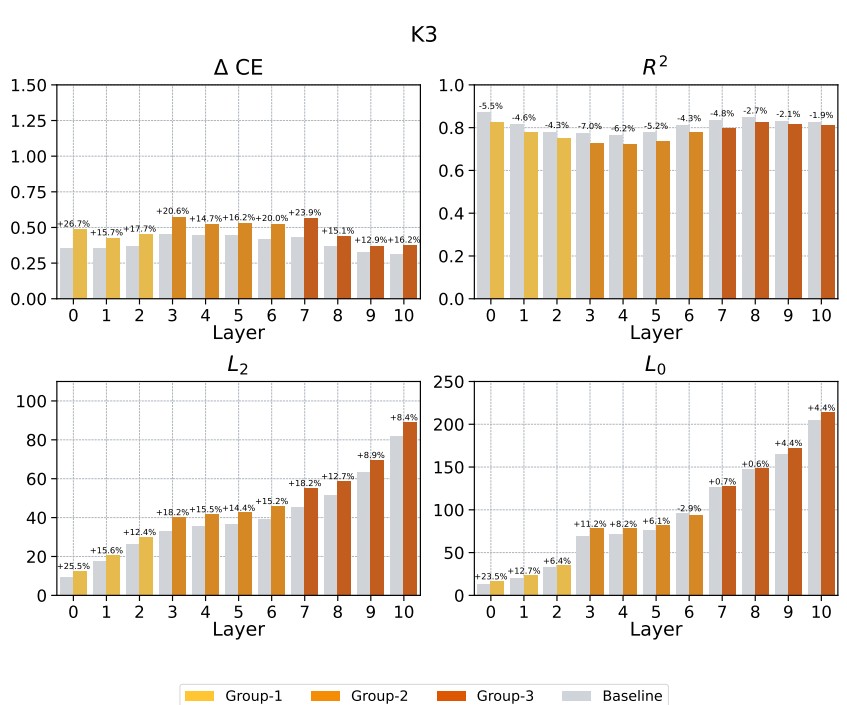

Figure 11: Per-layer $\Delta$CE, $R^2$, $L_2$ and $L_0$ with $k = 3$.

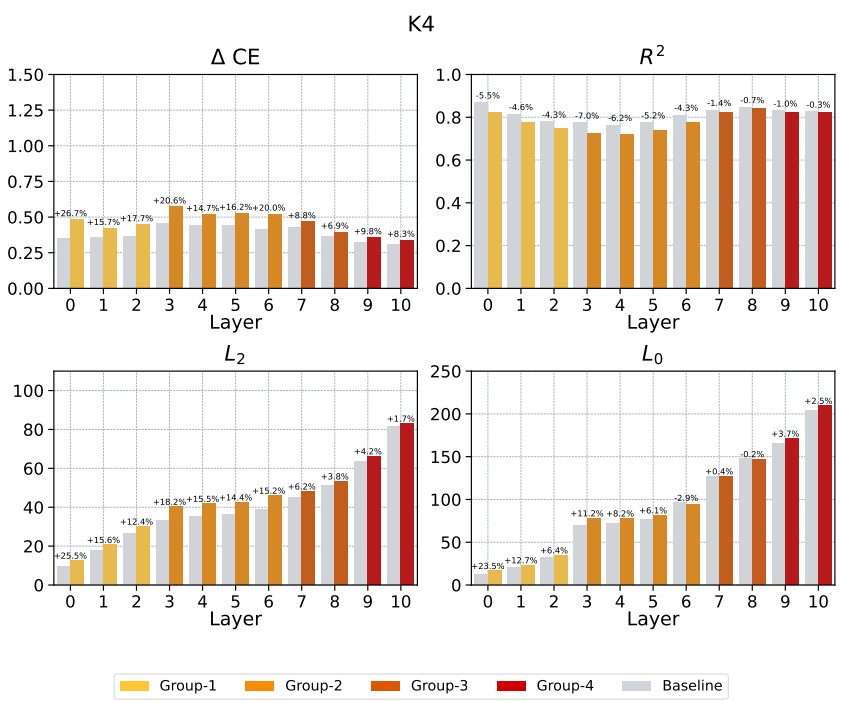

Figure 12: Per-layer $\Delta$CE, $R^2$, $L_2$ and $L_0$ with $k = 4$.

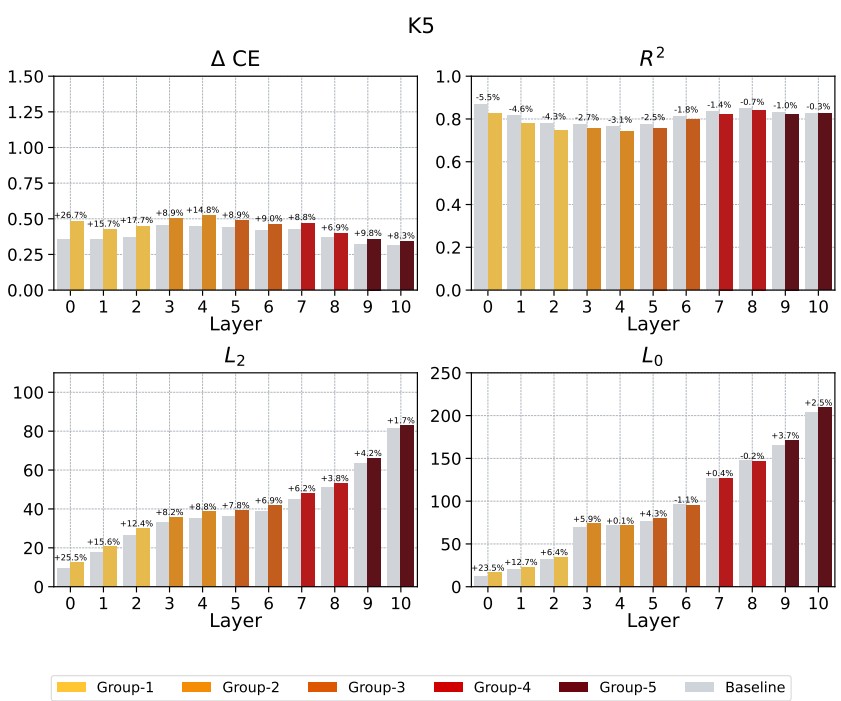

Figure 13: Per-layer $\Delta$CE, $R^2$, $L_2$ and $L_0$ with $k = 5$.

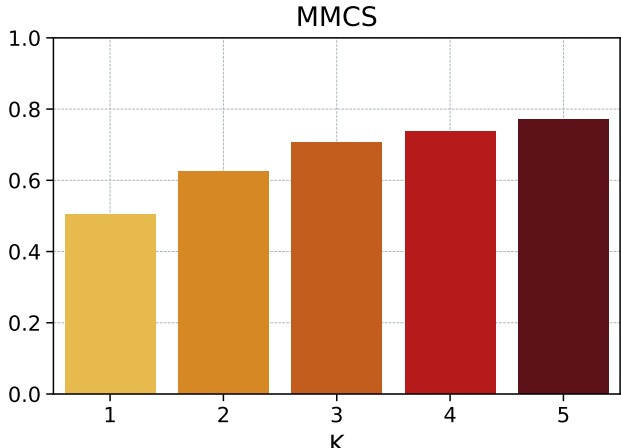

Figure 14: Per-layer $\Delta$CE, $R^2$, $L_2$ and $L_0$ with $k = 5$.

## D   DETAILED PER-LAYER MMCS PLOTS

The Maximum Mean Cosine Similarity (MMCS) Score is defined as:

$$\text{MMCS} = \frac{1}{d_{\text{sae}}} \sum_{\mathbf{u}} \max_{\mathbf{v}} \text{CosSim}(\mathbf{u}, \mathbf{v}) \tag{7}$$

where $\mathbf{u}$ and $\mathbf{v}$ are the columns of the $\text{SAE}_{j_k}$ and $\text{SAE}_i$ decoder matrices, respectively. Figure 14 shows the average MMCS, where the average is computed for a given $k$ by calculating the MMCS between $\text{SAE}_{j_k}$ and $\text{SAE}_s$ for every $1 \leq j \leq k$ and $s \in [j_k]$, then dividing by $L - 1$.

Figure 15 shows the per-layer MMCS of every $\text{SAE}_{j_k}$ for every $k$.

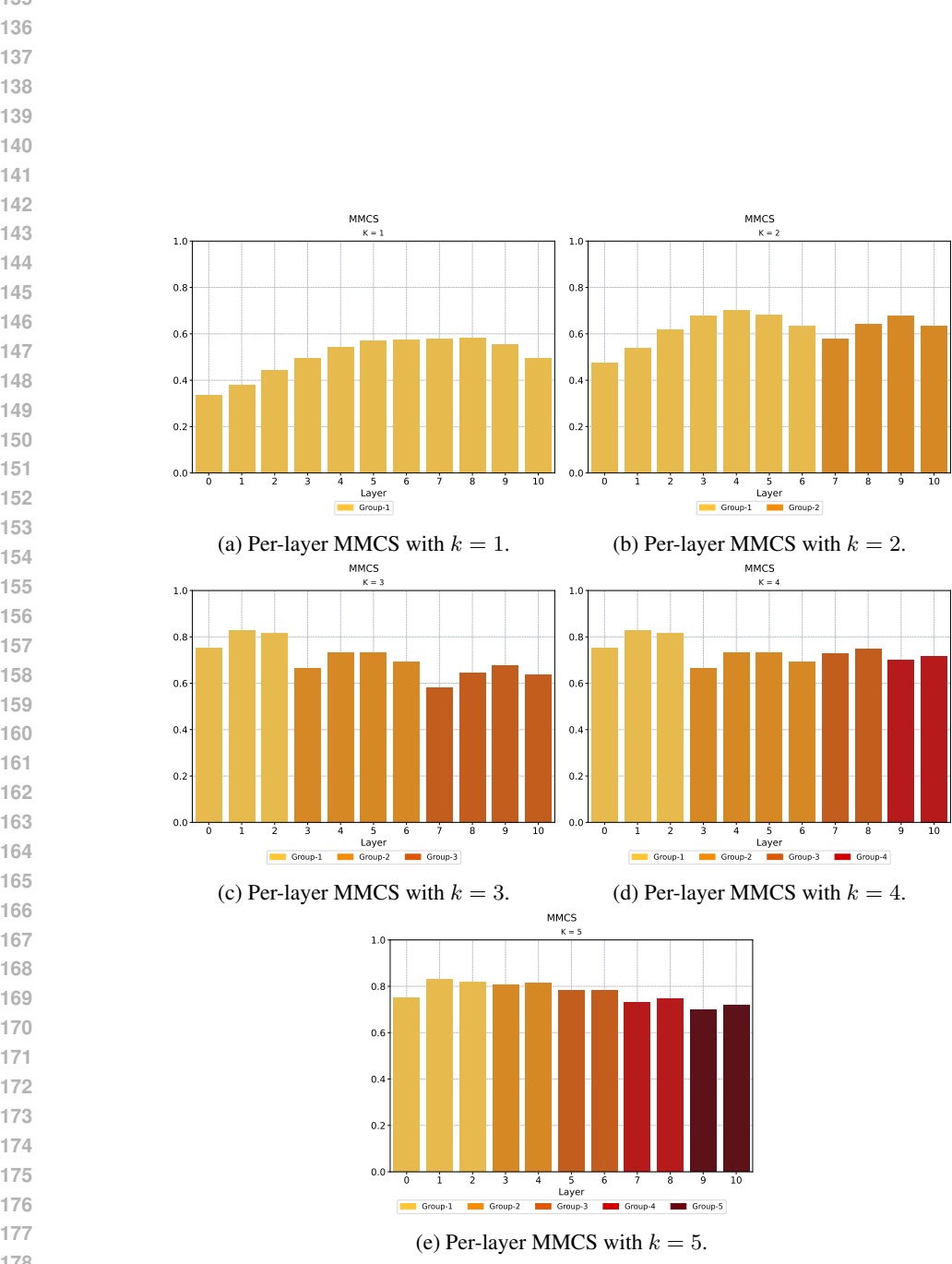

(a) Per-layer MMCS with $k = 1$.

(b) Per-layer MMCS with $k = 2$.

(c) Per-layer MMCS with $k = 3$.

(d) Per-layer MMCS with $k = 4$.

(e) Per-layer MMCS with $k = 5$.

Figure 15: Per-layer MMCS for every $k$.

# E    APPROXIMATE INDIRECT EFFECTS

In Equation 6 we reported the Indirect Effect (IE) (Pearl, 2022), which measures the importance of a feature with respect to a generic downstream task $\mathcal{T}$. To reduce the computational burden of estimating the IE with a single forward pass per feature, we employed two approximate methods: Attribution Patching (AtP) (Nanda, 2023; Syed et al., 2024) and Integrated Gradients (IG) (Sundararajan et al., 2017).

AtP (Nanda, 2023; Syed et al., 2024) employs a first-order Taylor expansion

$$\hat{\text{IE}}_{\text{AtP}}(m; \boldsymbol{f}_i; x_{\text{clean}}, x_{\text{corrupted}}) = \nabla m \big|_{\boldsymbol{f}_i = \boldsymbol{f}_{i;\text{clean}}} (\boldsymbol{f}_{i;\text{corrupted}} - \boldsymbol{f}_{i;\text{clean}}) \tag{8}$$

which estimates Equation 6 for every $\boldsymbol{f}_i$ in two forward passes and a single backward pass.

Integrated Gradients (Sundararajan et al., 2017) is a more expensive but more accurate approximation of Equation 6

$$\hat{\text{IE}}_{\text{IG}}(m; \boldsymbol{f}_i; x_{\text{clean}}, x_{\text{corrupted}}) = \left( \sum_{\alpha \in \Gamma} \nabla m \big|_{\alpha \boldsymbol{f}_{i;\text{clean}} + (1-\alpha) \boldsymbol{f}_{i;\text{corrupted}}} \right) (\boldsymbol{f}_{i;\text{corrupted}} - \boldsymbol{f}_{i;\text{clean}}) \tag{9}$$

where $\alpha$ ranges in an equally spaced set $\Gamma = \{0, \frac{1}{N}, ..., \frac{N-1}{N}\}$. In our experiments we have set $N = 10$.

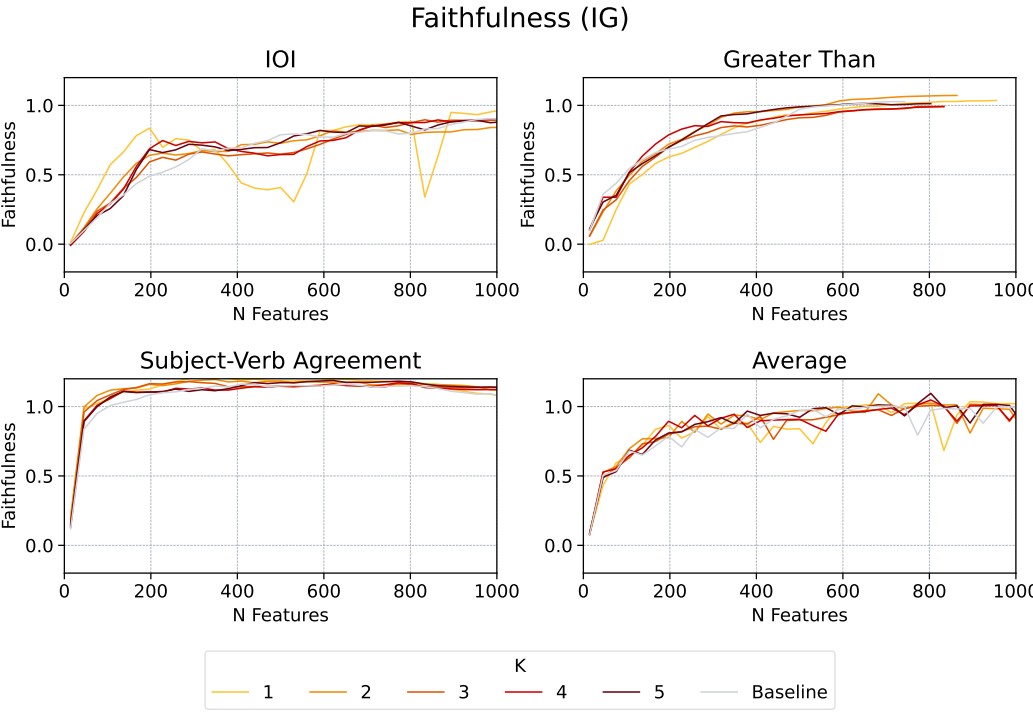

Figure 16: Average faithfulness (Marks et al., 2024) for every downstream task (IOI, Greater Than, Subject-Verb Agreement). The "Baseline" average is computed considering the performance obtained by $\text{SAE}_i$, $\forall i = 0, ..., 10$. The "Average" plot depicts the average over the three downstream tasks.

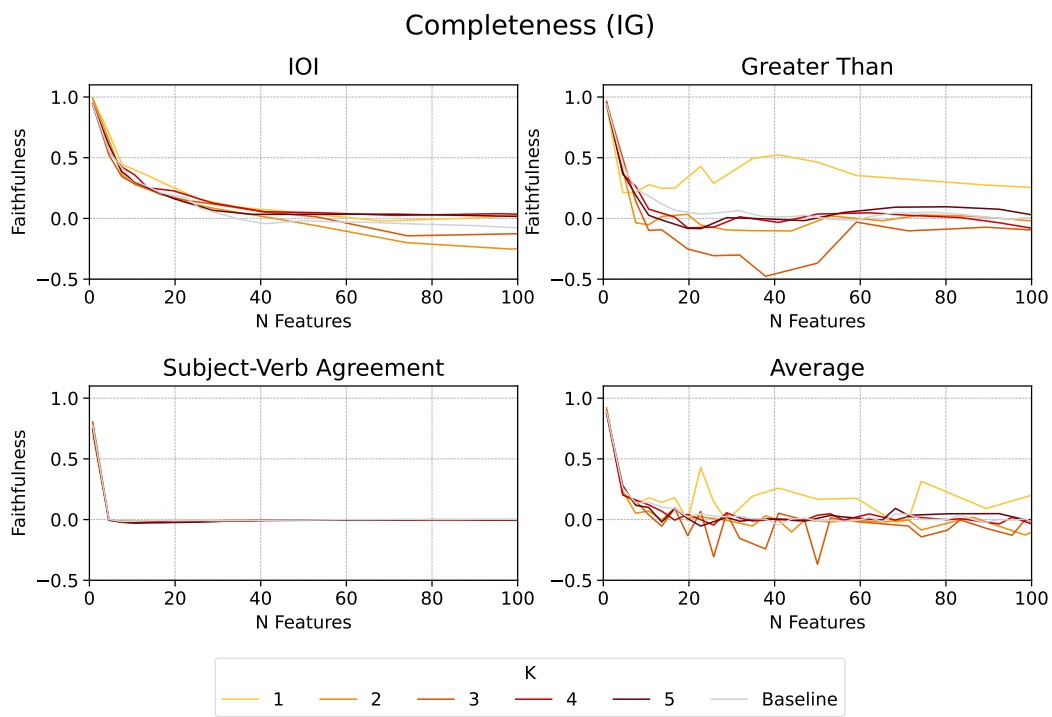

Figure 17: Average completeness for every downstream task (IOI, Greater Than, Subject-Verb Agreement). The "Baseline" average is computed considering the performance obtained by $SAE_i$, $\forall i = 0, ..., 10$. The "Average" plot depicts the average performance over the three downstream tasks.

## F    HUMAN INTERPRETABILITY SCORES

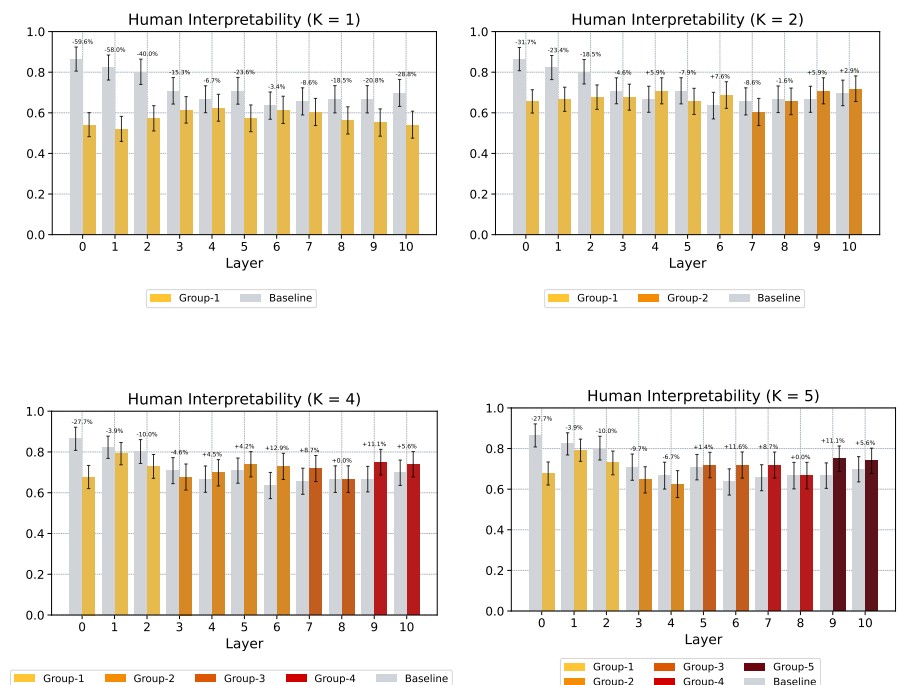

Figure 18: Human Interpretability Scores for $k \in \{1, 2, 4, 5\}$. The differences in the interpretability scores of features learned by the $\text{SAE}_{j_k}$ and the baseline $\text{SAE}_i$ are not statistically significant different from zero for most of the layers. Error bars shows one standard deviations of the scores differences, modeled as a Binomial distribution (Wasserman, 2010)

# G   SCALING GROUP SAEs

Results on Gemma-2 2b will be added here.

