# OpenReview forum: "Efficient Training of Sparse Autoencoders for Large Language Models via Layer Clustering"
_ICLR.cc/2025/Conference — ICLR 2025 Conference Withdrawn Submission_

### Official Review · Reviewer_mGhF · 2024-10-22

**Soundness:** 1
**Presentation:** 1
**Contribution:** 3
**Rating:** 6
**Confidence:** 4

**Summary:**

- When training sparse autoencoders on the residual stream, there is often one per layer. This is likely redundant, as the residual streams at two adjacent layers are often fairly similar
- The authors propose instead grouping layers by similarity of residual stream, using average angular distance, and training a single SAE for each group of layers.
- The authors claim that grouping is a substantial speedup. However, the text implies (but does not explicitly state) that all SAEs are trained on 1B tokens. This means that an SAE for a group of eg 3 layers trains on 3e9 activations, while training 3 SAEs, one for each layer, trains on 1e9 activations, for the same total compute. This means it is not a speedup. If the grouped SAE was instead trained on 0.33B tokens, or even randomly sampled one layer's residual for each token, this would be a speedup. This is a crucial detail and needs to be clarified
- The grouped SAEs are evaluated fairly carefully against the baseline of identically training an SAE per layer. The authors use a range of evaluations:
    - Standard metrics like L0, L2, CE Loss score
    - A circuit finding eval on several previously studied circuits. Authors use attribution patching to identify key SAE latents and calculate completeness and faithfulness. It does not seem to be stated whether there is a separate forward pass when ablating at each layer, or if all layers are ablated at at once.
    - A human interpretability study with 96 features. It is not specified whether this is 96 features per SAE, per SAE family, or total.
- The overall conclusion is that quality and performance was preserved, which seems reasonable for K=4 or K=5 at least.

**Strengths:**

- A fairly comprehensive set of evaluations is used, more than is common in such papers. I particularly liked the circuit based eval, modulo the concerns below
- It's a fairly simple, elegant idea that I hadn't seen done before, and which could be a simple drop-in replacement to significantly save the cost of training suites of residual SAEs
- Covered some key limitations clearly in the limitations section

**Weaknesses:**

- Numerous missing details, as discussed in the summary, which make it impossible to evaluate how impressive the results are
- Only studies a single model
- Others discussed below

**Questions:**

# Major Comments
*The following are things that, if addressed, might increase my score*
- My overall assessment here is that it's quite plausible that this technique works, just on priors, and would be valuable to people training suites of SAEs on every layer's residual stream like Gemma Scope. I like the spirit of this paper, and think it has the potential to be a solid paper, but that right now there's significant room for improvement and I am not currently convinced of the core thesis based on the evidence in the paper. **If rebuttals go well, I'm open to increasing my score to a 6 or possibly 8.**
- There are several crucial missing details in the paper, that significantly alter how the results are to be interpreted. I wouldn't be comfortable accepting the paper until these are clarified positively, or follow-up experiments are done which clarify, especially re the amount of compute used. (Note: I tried to check the paper fairly hard for these details, but I apologise if I missed them somewhere)
    - Is a layer group SAE trained on the same number of tokens as a normal SAE? (and so on significantly more activations, since there's one per layer in the group) If so, the claims of a speedup is false, this should take the same about of compute. There are minor benefits, like having fewer features to care about for eg probing, and fewer SAE parameters, but these are comparatively minor I think. To get a speedup, you need to run it on num_tokens / num_layers, for the same amount of training compute as a normal SAE (and less LLM running compute). It needs to be a fair comparison, as SAEs trained for longer generally perform better on evals, so the current evals can't be trusted.
    - What does "96 features" mean in the human interpretability study? Is it per SAE, per SAE family (ie a set of SAEs covering all 12 layers, either grouped or baseline), or 96 total? 96 isn't that much, so this significantly changes how much to trust the study.
    - With the circuit finding evaluations, when you do mean ablations to find completeness or faithfulness, do you do mean ablations to each layer one at a time, with a separate forwards pass per layer? Or to all layers at once?
        - If it's one pass per layer, then is it just averaged to form the faithfulness and completeness graphs?
        - If all layers are done at once, does N features mean per layer, or total?
        - Are you including error terms (ie the original act - reconstruction, as done in Marks et al), or not?
        - Including error terms makes comparisons harder - if an SAE is bad, its error terms are more informative, boosting performance
        - Meanwhile, if you don't include it and take ablations at every layer, I'm very skeptical that you could get the completeness results you do. In my experience, applying SAEs at every layer at once tanks model performance to the point of near randomness
- You say you use JumpReLU activations, but also that you use an L1 loss, unlike the JumpReLU paper's L0 loss + straight-through estimators. The threshold goes into a discrete function and so always has gradient zero and will never update unless straight-through estimators are used, suggesting that in this setup the thresholds are always zero? This is OK, it's just a ReLU SAE, but is misleading
- There are various things that make it messier to apply: it's not clear how many groups to make (especially on larger models!), I'm not convinced there's not a big performance degradation when done in a compute matched way, it's not clear how well these results generalise (in particular to the larger models where this is expensive enough to matter), etc. I know these are not realistic to fully address in the rebuttal, but here are some experiments that would strengthen the work:
    - Just doing the cosine sim analysis and layer grouping across several more models, including larger ones, and seeing how many groups are needed to get max angular distance below some threshold (eg whatever it took for 4 groups here). This should be fairly cheap, I think
    - Replicating these results on a larger model, eg Gemma 2 2B. Training a full suite would of course be fairly prohibitive, but eg finding an early pair of layers with low angular distance, and showing a compute matched grouped SAE performs comparably with an SAE per layer, would be compelling and should not be too expensive, especially with a low expansion factor, low number of tokens, and stopping execution of the LLM after the target layer.


# Minor Comments
*The following are unlikely to change my score, but are comments and suggestions that I hope will improve the paper, and I leave it up to the authors whether to implement them, either during rebuttals or after. No need to reply to all of them in the rebuttal*
- The distribution of residual streams in each layer is going to be different, in particular the norm and the mean will vary. I expect you could get notably better performance by pre-computing and subtracting the mean and dividing by the average norm (after mean-centering), to make layers comparable. This mean and scaling factor could even be made learnable parameters. I'd be curious to see this tried.
- Similarly, residual streams typically have significant non-zero mean (though I haven't investigated Pythia specifically) which makes cosine sim/angular distance harder to interpret, I'd be curious to see Figure 2 with mean-centered cosine sim (I don't expect major changes, but should be cleaner)
- I hypothesise that a layer grouped SAE trained on 1B tokens and acts from each layer may perform comparably to one trained on 1B tokens and a randomly chosen layer's act per token, since the acts are likely to be highly similar. This would also be a big SAE training compute reduction!
- The up to 6x speedup claim seems like an overclaim, it seems pretty clear to me that the K=1 setting performs badly enough to probably not be worth it. I'd say K=3 is the smallest that seems reasonable, so a 3x speedup (if compute matched)
- In Figure 3, I think it's pretty confusing to average your stats across all layers, especially things like L2 which are *very* different across layers. I would recommend either normalising (eg dividing by the baseline value), or ideally providing the per-layer info compactly. For example, a bar chart with layer on the x axis, and a group of 6 bars at each place (one for each K, and baseline). Or a line chart where the x axis is layer, and there's a line for each K and baseline. I appreciated these being included in the appendix, but this could be mentioned prominently in the main text
    - I also recommend displaying the raw change in cross-entropy loss, not the normalised CE score. Ablating the residual stream is extremely damaging, so normalised scores are always very high, making it hard to interpret
- Line 193: You say umpReLU is z * ReLU(z-theta), but it's actually z * H(z-theta), where H is the Heaviside function (1 if positive, 0 if negative)
- Figure 2: It would be good to clarify in the caption or line 209 that the key thing to look at is the main diagonal (and that it's about each layer to the adjacent one, not to itself!), that lower means closer, and that 0.5 means "totally perpendicular". I figured this out eventually, but it would help to clarify it
- I didn't find the discussion of MMCS on page 6 to be too informative. Without a baseline of comparing to another trained SAE on that layer, it's hard to really interpret what it should look like. I'd be fine with this being cut or moved to an appendix if you need the space
- Line 337: It would be good to clarify that you do integrated gradients by intervening and linearly interpolating the *activations* not *input tokens*. [Marks et al](https://arxiv.org/abs/2403.19647) does it your way, [Hanna et al](https://arxiv.org/abs/2403.17806) does it on input tokens (and input tokens is the standard method, though IMO less principled than your's)

---

> ### Author Response · Authors · 2024-11-22
>
> We thank the reviewer for his precious insights, and for the depth of the review. We believe we can address your concerns.
>
> # Group SAE Training
> No, we train one SAE per group with a number of tokens equal to T, with each layer in the group that receives an even amount of tokens equals to T/layers_in_group, and, as you mention, that’s where the computational speedup comes from. We fixed T=1B to maintain comparability with the amount of tokens used to train every single baseline SAE.
>
> # Human Interpretability
> The interpretability score of each group is the proportion of features that human annotators deem interpretable out of a set of 96 randomly chosen features per Group-SAE, averaged across every layer in the group. This means human annotators check 96 features per layer in each group (if we had 3 layers in a group we would check 96 * 3 features and take the proportion of interpretable features out of those as our human interpretability score).
>
> # Circuit Analysis
> When doing circuit analysis, the group SAE is “copied” for each layer it has been trained on. Then, ablations are done for each layer separately. This is motivated by the fact that each layer can require different feature of the same SAE to work effectively.
> As said in Marks et al, we include residuals in the reconstructions for circuit analysis. Removing errors doesn't allow to correctly perform this kind of evaluation.
>
> # JumpReLU
> We have indeed trained a JumpReLU SAE without normalizing the activations to have L2 norm equal to 1 during SAE training. We are in the process of training SAEs and Group-SAEs accounting for activations normalization, as requested in your minor comment.
>
> # Larger Models.
> We include angular values for Gemma-2 2B and 9B, and train a Group-SAE containing early layers. We are pleased to report the method scales, and thank you for your kind suggestion.
>
> # Normalizing Activations
> Thanks for the suggestion. We are now in the process of training baseline SAEs and Group-SAEs with centered and normalized activations such that they have expected L2 norm equal to 1 during SAE training, as specified in (cite JumpReLU paper). We have estimated the mean norm scaling factor on 2M tokens of the training set.
>
> # Training Procedure
> This is indeed our procedure: we randomly choose a layer’s act per token within a group and train the respective Group-SAE on exactly T tokens (each layers in a group thus receiving a total of T/layers_in_group tokens)
>
> # Speedup
> We agree that a 3x speedup is more reasonable given the decrease in performance, and will amend this is the revision.
>
> # Other fixes
> Following your suggestions, we made other fixes to our work which include:
> Figures 3 and 6 have been changed to their unaggregated form.
> We substituted CELS with Delta CE as it is more informative.
> Included an heuristic based on angular distances to choose K, the number of groups in larger models.
>
> Many thanks again for all your input.

---

> > ### Comment · Reviewer_mGhF · 2024-11-24
> >
> > > Group SAE Training
> >
> > To clarify, if say T=4000000 and num_layers_in_group=4, do you run the model on 1000000 tokens, on each token take all four layers, and train the residual stream on the 4*1000000 activation vectors? Or do you take 4000000 tokens, on each randomly (or uniformly?) choose one layer to take an activation vector from, to get 4000000 total activation vectors? (ie, will there be 1000000 forward passes or 4000000)
> >
> > I'll try to respond to the other comments later today, sorry for the delay

---

> > > ### Author Response · Authors · 2024-11-24
> > >
> > > Thank you for your response. It is the second indeed: we take 4000000 tokens, on each we randomly choose one layer to take an activation vector from. In this example, every baseline SAE is trained for 4000000 tokens for every of the four layer in the group, for a total of 16000000 tokens. This choice is motivated by the fact that we want the Group SAE to see the same number of tokens as the baseline SAE but having it able to reconstruct activations from multiple layers.

---

> ### Comment · Reviewer_mGhF · 2024-11-25
>
> Thank you for clarifying the training procedure here and in the main text, I think this is the natural training process and have no further concerns here.
>
> And thanks for improving figure 3, and clarifying the human interpretability study, I think this does a good job of quantifying the key metrics for exactly how much of a performance degradation this is.
>
> My remaining important concerns:
> - I'm not convinced by your faithfulness metric. The problem with keeping in error nodes is that the error nodes do far more on low quality SAEs - the opposite of what we want! This is fine when evaluating completeness (we just want to measure how much we lose when removing just the key features, so removing the error node is incorrect), but not for faithfulness. Faithfulness is measuring how well your features do at capturing the behaviour, and you are using this as a metric for the overall SAE quality, and so you need to compare SAEs. But on an extremely bad SAE, that eg always outputs zeros, you can get excellent faithfulness by ablating everything.
> - As far as I can tell, you also don't explicitly specify in the text how you aggregate the completeness and faithfulness per layer (it sounds like you just average?). It feels kinda hard to compare the average for each SAE group to the baseline - what if the problem is just inherently easier at later layers than earlier layers? Then even if the late groups are a degradation they may look better than baseline.
> - I still don't understand how you can train a JumpReLU SAE with an L1 loss
>
> Minor comments:
> - A key missing motivation for the technique is that the residual stream is resid_post[k] = resid_post[k-1] + attn_out[k] + mlp_out[k]. Each layer is an incremental change to this running sum, so of course SAEs should transfer
> - A key consideration re efficiency is how people are generating the activations, and what fraction of the compute budget this is. Your approach doesn't change the number of times the LLM is run to generate activations, so this part of the compute budget is unchanged.
> - A positive point is that the storage costs required for activations also go down by a significant factor, for those training SAEs without generating activations online.
>
> My overall take is that, while I still have methodological concerns about details of this paper and believe it could be substantially improved, the core story feels clearer - group SAEs were trained by a sensible method, evaluated via the standard metrics (sparsity-reconstruction plots, and human interpretability studies), and found to work with moderate degradation. On priors, I thought this technique was likely to work, as residual streams at nearby layers should be highly similar, so I believe this paper's evidence. And I think it is a useful method for SAE practitioners, at least in certain contexts where the compute spent training the SAE on activations or the storage space for activations is a big bottleneck. As such, I think this paper is a useful contribution to the literature, and I would consider using the technique in my work where appropriate.
>
> **As such, though I still have reservations, I am changing my score from a 3->6 and (marginally) believe this paper should be accepted**

---

> > ### Author Response · Authors · 2024-11-25
> >
> > Thank you for your thoughtful feedback and for pointing out these aspects for further clarification. Below, we address your remaining important concerns:
> >
> > ### Faithfulness Metric:
> > In our evaluation, we observed that faithfulness consistently rises from zero to one across all tasks. This suggests that when performing mean ablation of all SAE features while keeping the error node, the resulting task scores are very low. This indicates a weak contribution of the error node to the faithfulness score on the task. While we acknowledge that downstream evaluation remains an open challenge in SAE literature, we believe this behavior aligns with our expectations. Nevertheless, we recognize the importance of exploring further improvements to faithfulness evaluation in future work.
> >
> > ### Aggregation of Completeness and Faithfulness:
> > You are correct that we compute the average scores across layers and treat each GroupSAE as independent when applied to different layers. For additional clarity, we reference Marks et al. (2024), which provides a detailed explanation of the evaluation procedure. We agree that comparing the average for each SAE group to the baseline might mask inherent difficulties in different layers and will consider better alternatives in future work.
> >
> > ### Training with JumpReLU SAE:
> > Our SAEs were trained using an L0 loss, and we have corrected this typo in the revised version of the paper. We appreciate your understanding and attention to detail.
> >
> > We hope this clarifies your concerns and are grateful for your feedback, which has helped us improve the quality and clarity of the work.

---

> > > ### Comment · Reviewer_mGhF · 2024-11-25
> > >
> > > Thank you for the clarification, I maintain my score.

---

### Official Review · Reviewer_qGx2 · 2024-11-01

**Soundness:** 3
**Presentation:** 2
**Contribution:** 1
**Rating:** 3
**Confidence:** 4

**Summary:**

The paper proposes a more efficient method of training SAEs that groups similar layers together and trains a single SAE on each group. The grouping is done using aglommerative clustering between layers, where the layer to layer similarity is defined by the average angular distance between layer activations across 5M tokens. The authors compare their method with 5 different values of k (number of groups) against baseline sparse autoencoders on standard SAE metrics (L0, R^2, CE Loss score, and L0), and find that it is worse on these metrics. They also compare against circuit faithfulness and completeness metrics, where their method slightly improves on baselines.

**Strengths:**

- The clustering of layers to find the best groups for training a shared SAE on is interesting
- The evaluations of the interpretability and downstream performance of the SAEs are strong
- The problem is mostly well motivated: methods to reduce the computational bottlenecks of training large SAEs are important.

**Weaknesses:**

- The largest weakness of this work is that it is unclear how the proposed method works. Does it concatenate the layers? Does it train on an equivalent fraction of activations from each layer? Does it take in layer i and predict all of the other layers?
- It is also unclear what this method actually improves on or tells us about, besides simply reducing the total number of SAEs trained (L0s, losses, and interpretability are all significantly worse). Does it actually use less flops (since its plausible that training on more layers requires more flops)? Does it tell us something about how many features are shared across layers, and which layers share features?
- Because of the lack of experimental details, it is very unclear how this differs from prior work in this area: Residual Stream Analysis with Multi-Layer SAEs, https://arxiv.org/pdf/2409.04185
- The paper contains many typos and rushed writing.
- All bar plots should have a number showing the actual value of the bar, and error bars where they make sense.
- This work only examines residual layers, which is in some sense the “easiest” setting for this idea.

**Questions:**

- How does the circuit analysis work with the multiple layer SAEs? Can features in the “same” layer be connected? If not, is this might be an unfair comparison to baselines, because there are less feaatures overall to choose from?
- Why do you think the L2 of the lower Ks is higher?

---

> ### Author Response · Authors · 2024-11-22
>
> We appreciate your review and feedback on our submission and we would like to address your concerns by further discussing the weaknesses and questions raised.
>
> # Method
> We understand that the lack of clarity has negatively impacted our methodology and we would like to clarify all the doubts raised. In particular, our method trains exactly k SAEs, each of them on the same 1B tokens budget. To this end, every Group-SAE in a k-groups partition of layers is trained to reconstruct the activations of every layer within a group independently, without concatenating the activations. Moreover, every layer within a group processes exactly 1B/layers_in_group tokens exactly, thus making the overall tokens used to train the particular Group-SAE equal to 1B. We further stress out that a single Group-SAE trains a single encoder and a single decoder matrix per group.
>
> # Goal
> The primary goal of this work is to show that a single SAE can be applied on multiple layers, i.e. layer group. This produces a speedup in the training process from 2x to 6x, according to the formula in the paper. Although MMCS (which is included in the paper) can provide insight on how features are shared between Group and Baseline SAEs, we leave feature inspection and sharing as a future work as the main focus of the paper is to reduce the computational burden of training multiple SAEs.
>
> # Literature review
> To the best of our knowledge, at the time of the writing of this paper we found no similar works. As stated in the guidelines for reviewing, we’re not required to refer to papers that are more recent than 4 months. We thank you for the work you provided as we will cite it in the camera-ready of this work.
>
> # Typos
> Thank you for pointing out that there are typos in our work, we hope that we have fixed the errors in the new revision.
>
> # Plots
> Thank you for the suggestion: in our updated version we have updated all the plots by including values on top of bars and generally improved the overall readability of our plots.
>
> # Choice of Residual Stream
> This choice has been taken according to the limited amount of resources at our disposal. It is also motivated by the fact that the residual stream is the most popular position to train SAEs as done in other works (Gao et al. 2024, Ghilardi et al. 2024). We leave the training of SAEs on Attention or MLP activations for future works.
>
> We hope this clarification assists in understanding our contributions. Here are the answers to the questions:
>
> # Circuit Analysis
> The number of features to choose from is the same at each layer and features are not connected across layers that share the same group. So this is a fair comparison as for each layer we compare the baseline (the SAE trained only on that layer) with each group SAE that has been trained on it. The choice of keeping features separated is motivated by the fact that each layer can require different features of the group SAE to reconstruct its activation. We haven’t inspected this further as it’s not a goal of this work, but can be an interesting direction for future works.
>
> # L2 vs. K relationship
> When the number of groups k is lower, it means that a particular Group-SAE is trained on the activations of more layers, potentially far away from each others: take for example the setting in which k=1, i.e., we train a single SAE on the activations of all the layers of a model. In this particular setting, the single Group-SAE must be able to reconstruct the activations of different layers, potentially encoding different information and thus decomposing into different features, which instead must be shared between all layers. This can potentially harm the reconstruction, especially when one use low values of k (k=1 and k=2 in our experiments demonstrate this, even though the reconstruction values are quite similar)

---

> > ### Comment · Reviewer_qGx2 · 2024-11-25
> >
> > Thank you for your thorough response! I remain concerned about many of my and other reviewer points, especially novelty, impact, and overall clarity, so I will keep my score.

---

### Official Review · Reviewer_UJ4K · 2024-11-04

**Soundness:** 1
**Presentation:** 3
**Contribution:** 1
**Rating:** 6
**Confidence:** 4

**Summary:**

This paper proposes a method that will make suites of Sparse Autoencoders (SAEs) easier to use (as they will require fewer SAEs) and easier to train (large compute saving). The method is to train SAEs on the activations from contiguous blocks of layers in the model.

**Strengths:**

* The paper provides comprehensive analysis on the end artifact of their work, such as detailed circuit analysis evals, interpretability studies and accuracy metrics.

* The idea is easy to understand and the execution competently done.

* The results on the circuit analysis evals look strong, as there's barely any performance hit to using the strategy according to that eval.

**Weaknesses:**

The paper claims that there is a $(L-1) / k$ efficiency saving through using their method. But unless I misunderstand, since there are a fixed number of tokens used $T$ (1B in this case), and there will always be $LT$ total activations which all SAEs are trained on, the number of FLOPs used to train the SAEs will be **the same** using this method or not. Since language model activation saving can be amoritized (e.g.  https://arxiv.org/abs/2408.05147 or https://github.com/EleutherAI/sae) there is no theoretical benefit to saving LLM activation saving either.

The paper is titled as "Efficient Training of Sparse Autoencoders..." and hence unless I misunderstand some method, this paper does not achieve its goal and I cannot recommend it.

**Questions:**

Please answer whether my weakness is correct.

---

> ### Author Response · Authors · 2024-11-22
>
> We appreciate your review and feedback on our submission and we would like to address your concerns.
>
> Our method is not trained on LT tokens, instead the budget per Group-SAE is fixed at T=1B tokens (the same for the baseline SAEs), and the Group-SAE is trained on T/layers_in_group for every layer within a particular group. To be more precise: given the pythia-160m model with 12 layers used in all our experiments, suppose k=3 (i.e., we cluster the layers into 3 groups) and suppose, for the sake of simplicity, that every group contains the same number of layers (without diminishing our claims related to grouping), i.e., in this example every group has associated 4 layers.
> Baseline SAEs are trained so that every SAEs reconstruct 1B tokens activations at every layer, i.e., a total of 12 SAEs * 1B tokens. Our Group-SAE instead trains k=3 SAEs, each of them reconstructing the activations from a group of layers, with 1B tokens per group, i.e., every layer in a group processes exactly 1B/num_layers_in_group=¼=0.25B tokens. The total number of tokens is k SAEs * 1B, which in our example is 3B, yielding the L/k speedup claimed in the paper (with a -1 since we do not consider the last layer).
>
> We hope that our explanations reduce the ambiguity of the current presentation towards a better understanding of our methodology.

---

> > ### Comment · Reviewer_UJ4K · 2024-11-25
> > **I updated my score to a 6**
> >
> > Thank you for clarifying. I updated to a 6. I think this is an interesting speedup when training large numbers of SAEs. I think that the most important SAE training problem is decreasing FVU while keeping the same sparsity at *one* site, rather than efficiently training lots of SAEs. Then benefits of your approach there are much lower, so my score is only a 6. Thanks for the comprehensive evals!

---

### Official Review · Reviewer_jU8Z · 2024-11-04

**Soundness:** 2
**Presentation:** 2
**Contribution:** 2
**Rating:** 3
**Confidence:** 3

**Summary:**

This paper builds on the recent line of work that relies on sparse autoencoders (SAEs) to address the interpretability of large language models (LLMs). In particular, SAEs aim to decompose LLM activations in a layer as a sparse combination of a large number of (interpretable) features. However, prior works require training one SAE per LLM layer (component), resulting in a large number of parameters and prohibitively high compute cost needed to obtain good quality SAEs to understand the inner workings of the LLM.

This paper leverages similarities among consecutive layers in an LLM to reduce the training cost for SAEs. The paper proposes to cluster LLM layers in $k$ groups and then train one SAE for each group of layers. Based on the reconstruction error of original representations; downstream performance on tasks focused on indirect object identification, greater than relationship, and subject-verb agreement; and human evaluations, the paper argues that the proposed approach results in good quality SAEs for Pythia 160M LLM.

**Strengths:**

1) The paper focuses on important and timely questions related to the interpretability of LLMs.
2) The proposed method successfully improves the training efficiency of SAEs for LLMs by grouping similar layers.
3) Empirical evaluation based on both reconstruction error and downstream performance showcases the utility of the proposed approach.

**Weaknesses:**

1) The main weakness of the paper is its limited technical novelty and contributions. The reviewer believes that the proposed approach of grouping multiple similar layers and training one SAE per group does not constitute a significant contribution to the field. Furthermore, the empirical evaluation in the paper is restricted to a small language model (Pythia 160M) and focuses on very simplistic tasks. This does provide strong evidence of the value of the proposed method for realistic settings involving LLMs.

2) There is a significant scope for improving the presentation of the paper. Many design choices in the paper are not well justified (see the Questions section below).

3) The authors build on many recent prior works. The reviewer believes that the authors can provide a more comprehensive background of some of these works to make the paper self-contained. It would be helpful for the reader to know how SAEs can be utilized for a particular application while studying LLMs.

**Questions:**

1) Please consider introducing dimensions of various vectors and matrices in Sections 2.1 and 2.2. Also, what is the relationship between $d$ and $dmodel$ (Line 112-113)? It appears that $d = dmodel$?

2) Please formally define the term "residual stream".

3) In Section 3.1, what is the justification/motivation for using *JumpReLU*?

4) As for the hierarchical clustering strategy described in Lines 212 - 220, is it clear that one will only put consecutive layers in one group? Can non-consecutive layers be clustered into a single cluster? If yes, is this desirable?

5) Figure 3 presents multiple metrics, CE loss, $R^2$, $L_2$, $L_1$. Out of these, which one is more important? Also, looking at some of the figures, the difference in the metric value for different $k$ is very small. What is the significance of this small difference?

6) In Figure 7, Human interpretability scores appear to be *non-monotonic* with respect to $k$. Could authors comment on this?

---

> ### Author Response · Authors · 2024-11-22
>
> We appreciate your feedback on our submission and would like to address a few points regarding the highlighted weaknesses:
>
> # Model size
> The model has been chosen according to the computational resources
> at our disposal. Choosing a bigger one would have made costs prohibitive, considering all the experiments that we have done. We plan to address this limitation by including a single group trained on a limited number of layers from a bigger model, such as Gemma-2b.
>
> # Novelty
> To the best of our knowledge, at the time we wrote the paper, no other works focused on the same topic, especially in the context of increasing SAE training efficiency.
>
> We hope this clarification assists in understanding our contributions. Here are the answers to the questions:
>
> # Vector dimensions
> We have added dimensions of vectors and matrices used in the experiments. Yes d and d_model corresponds.
>
> # Residual stream
> The residual stream is a well known concept in interpretability, and represents the activations of the model after summing the attention and MLP at each layer. In our work we refer to the residual stream after the MLP layer contribution. We have also added a formal definition in the new revision of the paper.
>
> # JumpReLU
> JumpReLU is, to the best of our knowledge, the SoTA approach regarding SAE training, as it obtains the best trade-off between sparsity (measured by the L0 norm) and reconstruction (measured by the L2 between the activations and their reconstructions). We hypothesize our method to transfer also to other architectures such as TopK, but for the limited computational resources, we leave this analysis as a future work.
>
> # Consecutive Layers in Clustering
> Yes theoretically two non-consecutive layers can be clustered together. However, in the experiments we have conducted on bigger models following the submission of the paper (which we have now added to the last revision), we found no empirical evidence on non-consecutive layers clustered together.
>
> # Performance Metrics
> Figure 3 reported all the most popular metrics to evaluate reconstruction performance of SAEs. There are no clear guidelines on which one is better in the literature so we report all of them to have a holistic view of the reconstruction performance. Regarding the second point, having small differences in the metrics means that group SAEs achieve performance similar to the baseline SAE in terms of reconstruction, thus validating our approach.
>
> # Human Interpretability
> There is no precise intuition about differences in human interpretability scores between different layers. Nevertheless, we added error bars to the scores to show that these small differences are not statistically significant.

---

### Official Review · Reviewer_Sre6 · 2024-11-08

**Soundness:** 2
**Presentation:** 3
**Contribution:** 1
**Rating:** 3
**Confidence:** 5

**Summary:**

This work proposes to reduce the computational overhead of SAEs: instead of training a separate SAE for each layer, it groups the layers to several groups of adjacent layers and learns an SAE for each group.

**Strengths:**

Improving LLM interpretability is an important topic. The proposed method of speeding up SAEs is straightforward and easy to understand.

**Weaknesses:**

Overall I feel that the results presented in this work are quite obvious and expected, and I do not see a large contribution to the community.
1. The novelty of this work is limited. It seems to be an obvious choice for one to learn an SAE for each group of adjacent layers.
2. Based on the experimental results (on a 12 layer 160M model), the speed up provided by the method is limited. The speed up also always comes with a drop of the quality of the model. Based on Figure 3 and Figure 4, the drop seems to be almost linear to k. This is quite expected with any types of "simple" speed up such as down sampling and grouping (like this paper suggested).

**Questions:**

See above.

---

> ### Author Response · Authors · 2024-11-22
>
> We appreciate your feedback on our submission and would like to address a few points regarding novelty, speedup, and performance:
>
> # Limited Novelty
> We respectfully disagree with the assessment that our method is “obvious.” To the best of our knowledge, grouping adjacent layers in Sparse Autoencoders for interpretability has not been attempted previously, and we present a detailed rationale for this choice.
>
> # Speedup
> Our method can achieve a speedup of between 2x and 6x for Pythia-160m, for example, with k=2, we only train two SAEs (one for each group), and for 1B tokens each (with each layer in the group being trained on 1B/number_of_layers_in_group tokens). This costs us 2B tokens instead of 11B tokens, which could be regarded as a significant speedup. Although this model shows an important drop in reconstruction performance, it is still useful for downstream and interpretability evaluation, which makes it useful in specific use settings.
>
> # Limited Performance Drop
> While any acceleration technique may introduce minor trade-offs, the drop in performance we observe is minimal, and interpretability remains high. Despite a near-linear relation to k, model quality is maintained at high levels. Moreover, we decided to substitute Figures 3 and 6 with their non-aggregated versions to show the differences for all metrics at each layer. We designed the method to balance speed and interpretability effectively, which we hope addresses the concern about “limited” speedup.
>
> # Interpretability Contribution
> We acknowledge that interpretability in LLMs is a challenging field, and our approach offers a novel method for balancing speed and interpretability, contributing to the existing interpretability tools.
> Thank you again for your insights, and we hope this clarification assists in understanding our contributions. If there are any additional points that would strengthen our submission, we would be glad to address them.

---

### Note · Authors · 2024-11-26

**Comment:**

Dear ICLR 2025 Committee,

We would like to formally withdraw our submission. This decision has been made after careful deliberation based on the feedback received during the review process. While we greatly value the constructive insights provided by the reviewers, we believe it is in the best interest of our work and for the community to address the concerns raised in more depth and conduct additional experiments before resubmitting to a future venue.

We sincerely thank the reviewers and the area chair for their time, effort, and feedback, which have been immensely helpful in guiding the next steps for our research.

Best regards,
The authors

**Withdrawal Confirmation:**

I have read and agree with the venue's withdrawal policy on behalf of myself and my co-authors.